# Nash Equilibria in Games with Playerwise Concave Coupling Constraints: Existence and Computation

**Philip Jordan** [1]   **Maryam Kamgarpour** [1]

## Abstract

We study the existence and computation of Nash equilibria in concave games where the players' admissible strategies are subject to shared coupling constraints. Under playerwise concavity of constraints, we prove existence of Nash equilibria. Our proof leverages topological fixed point theory and novel structural insights into the contractibility of feasible sets, and relaxes strong assumptions for existence in prior work. Having established existence, we address the question of whether in the presence of coupling constraints, playerwise independent learning dynamics have convergence guarantees. We address this positively for the class of potential games by designing a convergent algorithm. To account for the possibly nonconvex feasible region, we employ a log barrier regularized gradient ascent with adaptive stepsizes. Starting from an initial feasible strategy profile and under exact gradient feedback, the proposed method converges to an $\epsilon$-approximate constrained Nash equilibrium within $\mathcal{O}(\epsilon^{-3})$ iterations.

## 1. Introduction

Optimization problems involving multiple self-interested agents with coupled objectives frequently arise in applications across economics (Krawczyk & Uryasev, 2000), computer science (Menache & Ozdaglar, 2011; Sahiner et al., 2022), and robotics (Wang et al., 2019). While in standard game theoretic frameworks, the players' feasible action sets are decoupled, in many interactive tasks, players are restricted to optimizing over a set of admissible strategies that are influenced by the choices of others. This game setting is captured by introducing coupling constraints, that is, constraints which depend on the players' joint strategy profile. For instance, in economic modeling, such constraint games have been applied to analyze equilibria under resource limitations and market constraints (Arrow & Debreu, 1954). In other applications, players may be subject to joint safety requirements such as collision avoidance in multi-robot control (Gu et al., 2023), or operating within a shared budget, such as limited bandwidth in communication networks (Lasaulce & Tembine, 2011). In machine learning, convex-concave min-max problems with shared coupling constraints have recently appeared in the analysis of Generative Adversarial Networks (Sahiner et al., 2022).

Despite their high relevance, several questions about the existence and computation of equilibria in games with coupling constraints remain open. The central challenge is that coupling constraints destroy the separability of players' feasible response sets, significantly complicating existence and computation of equilibria.

In this paper, we consider games with playerwise concave coupling constraints and playerwise concave utilities. We prove the existence of equilibria and design an algorithm that provably converges to an equilibrium. To place our contributions, we discuss the state-of-the-art on existence and computation of equilibria for the considered class of games.

**Existence**   Shortly after Nash's foundational work on the existence of mixed-strategy equilibria in finite games with decoupled action sets (Nash, 1950), Debreu introduced games with coupling constraints (Debreu, 1952). In his formulation, each player may face a distinct coupling constraint, a setting we refer to as *non-shared coupling constraints*. A constrained Nash equilibrium is then defined as a strategy profile from which no player has an incentive to unilaterally deviate within her feasible set[1].

For non-shared constraints, Debreu established existence of constrained Nash equilibria under playerwise quasi-concavity of utilities. However, his proof is based on the assumption that each player's feasible set is nonempty regardless of the strategies chosen by others. This assumption

---

[1]EPFL, Switzerland. Correspondence to: Philip Jordan <philip.jordan@epfl.ch>.

*Proceedings of the 43rd International Conference on Machine Learning*, Seoul, South Korea. PMLR 306, 2026. Copyright 2026 by the author(s).

---

[1]We remark that this setting is also commonly known under the term generalized Nash equilibrium problem, particularly in the variational inequality literature (Facchinei et al., 2007).

is restrictive: in many practical scenarios, such as safety or threshold constraints on players' strategies, violations induced by a subset of players may leave others with no feasible response. Building on Debreu's framework, a similar nonemptiness assumption has appeared in the context of constrained Markov games under the name *strong Slater condition* (Altman & Shwartz, 2000); here as well, this condition may be violated in practical applications such as safety constraints in autonomous driving when violations are caused by collisions among a subset of players.

For *shared* coupling constraints, where all players face the same constraints, Rosen showed that constrained Nash equilibria exist under playerwise concavity of utilities (Rosen, 1965). While this result did not require the restrictive assumption of Debreu (1952) on each player's feasible response set, it relied on the assumption that the feasible region is jointly convex.

The joint convexity assumption limits the applicability of the result: even simple bilinear coupling constraints lead to non-jointly convex feasible regions.

The above results on existence of equilibria in games with coupling constraints raise the question of whether the equilibrium existence can be guaranteed for games with shared coupling constraints under weaker conditions on the feasible response sets or on the the joint strategy space.

We resolve this question for the class of games with playerwise concave constraints, a relaxation of joint concavity that allows constraints to depend on other players' actions in a nonconcave manner. Playerwise concavity is a natural assumption: just as playerwise concavity of utilities suffices for equilibrium existence in unconstrained games, playerwise concavity of constraints ensures convexity of individual feasible sets without imposing restrictive joint structure.

**Computation**  While existence results are fundamental for Nash equilibria to be plausible outcomes of multi-agent interactions, practical relevance also requires computational tractability. Without further assumptions on the game structure, computing Nash equilibria is intractable even in two-player games (Daskalakis et al., 2009; Papadimitriou et al., 2023). Consequently, positive algorithmic results exist for structured classes of games. In particular, in the absence of coupling constraints, the class of potential games (Heliou et al., 2017; Anagnostides et al., 2022) and zero-sum games (Mokhtari et al., 2020; Mertikopoulos et al., 2019) admit efficient algorithms and well-understood learning dynamics.

By contrast, in the presence of coupling constraints, the computational landscape remains far less understood, even under favorable structural assumptions such as in zero-sum

or potential games.

It has been shown that convex-concave zero-sum two-player games with non-shared coupling constraints are computationally intractable (Daskalakis et al., 2021; Bernasconi et al., 2024); a stark contrast to their unconstrained counterparts. Tractability can be recovered when constraints are shared and the feasible region is jointly convex, where the problem can be address through well-studied variational inequality framework (Facchinei & Pang, 2003).

Beyond joint convexity, however, little is known. In fact, even when constraints are shared and feasible sets are playerwise rather than jointly convex, finding a feasible strategy is NP-hard (Witsenhausen, 1986). This rules out algorithmic approaches that rely on first solving feasibility and motivates a different perspective: Starting from an initial feasible strategy, can players' learning dynamics converge to a constrained Nash equilibrium?

We pursue this direction for potential games with playerwise concave utilities and shared playerwise concave coupling constraints. In the absence of coupling constraints, independent learning dynamics is known to converge to Nash equilibria in such games (Anagnostides et al., 2022). We show that even in the presence of shared coupling constraints, there exists a playerwise decoupled learning algorithm that converges to an equilibrium.

Summarizing the above, our contributions are as follows.

**Contributions**

1. For the class of games with playerwise convex feasible regions, we establish the existence of constrained Nash equilibria without requiring nonemptiness of feasible response sets (Debreu, 1952) or the joint convexity of the feasible region (Rosen, 1965).

   A key technical contribution of our work is a novel proof technique for analysing the topological structure of the feasible region. This allows us to build on tools from topological fixed point theory, specifically the works of Eilenberg & Montgomery (1946), and Begle (1950), which extend fixed point results from convex to more general, contractible domains.

2. Restricting the above class to potential games with coupling constraints, we develop a playerwise decoupled learning algorithm with provable convergence. In our approach, players simultaneously update their strategies using only gradient feedback on their own utility and the shared constraints, without requiring coordination or communication.

   Our algorithmic novelty is to extend a log barrier interior point method (Frisch, 1955) from single-player constrained optimization (Megiddo, 1989; Hinder &

Ye, 2024) to the multi-player game setting. Starting from an initial feasible strategy, and by carefully selecting adaptive learning rates, we guarantee convergence to an $\epsilon$-approximate constrained Nash equilibrium at an $\mathcal{O}(\epsilon^{-3})$ rate.

3. We empirically demonstrate convergence of the proposed algorithm in a cooperative game with a nonconvex feasible region and in a network routing game with coupled link capacity constraints.

**Paper outline** We begin by introducing the constrained game setting in Section 2. In Section 3, we provide our existence result. Section 4 then focuses on learning dynamics in constrained potential games. Lastly, we describe our simulations in Section 5 and conclude the paper with Section 6.

## 2. Preliminaries

**Notation** For a finite set $X$, we denote the probability simplex over $X$ by $\Delta(X)$. For $n \in \mathbb{N}$, we use the notation $[n] := \{1, \ldots, n\}$. For $x \in \mathbb{R}^n$, let $\|x\|$ denote the $\ell_2$-norm.

**Game setting** In this paper, we study static games with continuous strategy spaces. Let $[m]$ be the set of players. For each $i \in [m]$, $\mathcal{X}_i \subset \mathbb{R}^{d_i}$ is a compact convex set of available strategies. Joint strategy profiles are denoted by $x = (x_1, \ldots, x_m) \in \mathcal{X} := \times_{i \in [m]} \mathcal{X}_i$. Upon playing strategy $x \in \mathcal{X}$, each player $i$ receives utility $u_i(x)$ where $u_i : \mathcal{X} \to \mathbb{R}$ is continuous. Shared coupling constraints are introduced through a set of functions and thresholds $\{(c_j, \alpha_j)\}_{j \in [b]}$ where for each $j \in [b]$, $\alpha_j \in \mathbb{R}$, and $c_j : \mathcal{X} \to \mathbb{R}$ is a continuous function. Players are assumed to be utility maximizers subject to lower bounds on the constraint functions. The set of strategy profiles considered feasible is given by

$$\mathcal{C} := \left\{ x \in \mathcal{X} \mid \forall j \in [b], \ c_j(x) \geq \alpha_j \right\}.$$

For joint strategies of all players other than $i$, we use the notation $x_{-i} := (x_1, \ldots, x_{i-1}, x_{i+1}, \ldots, x_m) \in \mathcal{X}_{-i} := \times_{j \in [m] \setminus \{i\}} \mathcal{X}_j$. For $i \in [m]$ and $x_{-i} \in \mathcal{X}_{-i}$, let

$$\mathcal{C}_i(x_{-i}) := \{x_i \in \mathcal{X}_i \mid (x_i, x_{-i}) \in \mathcal{C}\}$$

denote the feasible response set of player $i$.

**Definition 2.1.** A *constrained Nash equilibrium* is a feasible strategy $x \in \mathcal{C}$ such that for all $i \in [m]$ and $x_i' \in \mathcal{C}_i(x_{-i})$, we have $u_i(x_i', x_{-i}) \leq u_i(x)$.

Playerwise concavity of utilities is commonly assumed for proving existence of Nash equilibria even without constraints. We naturally make an equivalent playerwise concavity assumption on the constraints.

**Assumption 2.2.** For all $i \in [m]$, $x_{-i} \in \mathcal{X}_{-i}$, and $j \in [b]$, $x_i \mapsto u_i(x_i, x_{-i})$ and $x_i \mapsto c_j(x_i, x_{-i})$ are concave.

**Remark** Restricting our attention to *shared constraints* is motivated by the fact that in the absence of additional assumptions—such as those imposed in Debreu (1952)—a constrained Nash equilibrium need not exist; see Appendix B for an example.

## 3. Existence of Constrained Nash Equilibria

In this section, we establish our existence result for constrained Nash equilibria. To this end, we introduce an additional assumption on the feasible set, which, for playerwise concave constraints, generalizes the existence conditions of prior work. Our key insight is that the feasible response set must have a well-behaved boundary (see Example 3.2); this can be achieved with conditions less stringent than prior work, as shown through Example 3.3.

### 3.1. Structure of the Feasible Region

We first introduce the following assumption, which ensures that at every boundary of the feasible region $\mathcal{C}$, for each player $i \in [m]$, there exists a uniform descent direction for all constraints.

**Assumption 3.1.** Let $\mathcal{J}(x) := \{j \in [b] \mid c_j(x) = \alpha_j\}$ be the set of active constraints for a feasible strategy $x \in \mathcal{C}$. We assme that for all $x \in \mathcal{C}$ and $i \in [m]$, there exists $\delta_i \in \mathbb{R}^{d_i}$ with $x_i + \delta_i \in \mathcal{X}_i$ and $\|\delta_i\| \leq 1$, such that for all $j \in \mathcal{J}(x)$, we have $\langle \delta_i, \nabla_{x_i} c_j(x) \rangle > 0$.

The above condition is a playerwise variant of the Mangasarian-Fromovitz constraint qualification condition (Mangasarian & Fromovitz, 1967). This condition is widely employed in single-player constrained optimization, and similar assumptions are commonly made in constrained game formulations (Dreves et al., 2011). In order to understand the implications of Assumption 3.1, in the following example, we first illustrate a case in which it does not hold.

**Example 3.2.** Consider a two-player game with continuous strategy spaces $\mathcal{X}_1 = \mathcal{X}_2 = [0, 1]$ and feasible set $\mathcal{C}^{(1)}$ defined by $c_1(x_1, x_2) := (x_1 - \frac{1}{2})(x_2 - \frac{1}{2})$ with threshold $\alpha_1 = 0$; illustrated on the left-hand side of Figure 1. Note that at the point $\hat{x} = (\frac{1}{2}, \frac{1}{2}) \in \mathcal{C}$, we have $\nabla_{x_1} c_1(\hat{x}) = \nabla_{x_2} c_1(\hat{x}) = 0$, implying that Assumption 3.1 is violated.

The main challenge in Example 3.2 is that at $\hat{x} = (\frac{1}{2}, \frac{1}{2})$, the set-valued feasible response mappings $x_1 \mapsto \mathcal{C}_2(x_1)$ and $x_2 \mapsto \mathcal{C}_1(x_2)$ are upper but not lower-semicontinuous (see Appendix C for definitions). This leads to discontinuity of the best-response maps, and as a result, fixed-point arguments for continuous self-maps no longer apply to guarantee existence of an equilibrium.

Next, in order to demonstrate the generality of our conditions, we present an example of a constrained game whose feasible region $\mathcal{C}^{(2)}$ satisfies both our Assumptions 2.2

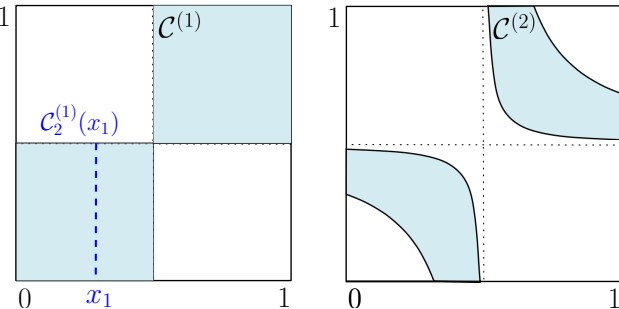

*Figure 1.* Illustration of the feasible sets $\mathcal{C}^{(1)}$ and $\mathcal{C}^{(2)}$ described in Example 3.2 and Example 3.3, respectively.

and 3.1, but does not fulfill the joint convexity or nonempty feasible response requirements imposed by previous existence results. Non-jointly convex feasible regions of this type arise naturally in mixed strategy extensions of matrix games (as defined in Example 4.2).

**Example 3.3.** Let again $\mathcal{X}_1 = \mathcal{X}_2 = [0, 1]$ and consider the feasible set $\mathcal{C}^{(2)}$ defined by two constraints, $c_1(x_1, x_2) :=$ $(x_1 - \frac{1}{2})(x_2 - \frac{1}{2})$ and $c_2(x_1, x_2) := -(x_1 - \frac{1}{2})(x_2 - \frac{1}{2})$, together with thresholds fixed, for concreteness, at $\alpha_1 = \frac{1}{111}, \alpha_2 = -\frac{1}{11}$, as shown on the right-hand side of Figure 3.3. The utility structure is irrelevant for this example. The resulting feasible set $\mathcal{C}^{(2)}$ is illustrated as the blue region on the right-hand side of Figure 1.

Observe that the constraints satisfy Assumption 2.2, as fixing $x_1$ yields a linear function in $x_2$, and vice versa. Moreover, Assumption 3.1 holds, as $|\mathcal{J}(x)| = 1$ for all $x \in \mathcal{C}^{(2)}$ and neither $c_1$ nor $c_2$ has any critical points on the boundary of $\mathcal{C}^{(2)}$. Furthermore, note that $\mathcal{C}^{(2)}$ is not jointly convex, hence Rosen (1965)'s result is not applicable. Moreover, for any $x \in \mathcal{X}$ with $x_1 = 0.5$, we have $\mathcal{C}_2^{(2)}(x_1) = \emptyset$. Hence, results relying on the nonemptiness of feasible responses such as Debreu (1952) are not applicable either.

Having motivated and clarified our setting, we address the existence of Nash equilibria for the class of constrained games captured by Assumptions 2.2 and 3.1.

### 3.2. Existence Result

**Theorem 3.4.** *Let Assumptions 2.2 and 3.1 hold, let $\mathcal{C} \neq \emptyset$. Then there exists a constrained Nash equilibrium $x^\star \in \mathcal{X}$.*

The full proof of Theorem 3.4 is given in Appendix D; here we sketch the main elements.

A key challenge in our setting is the nonconvexity of the feasible region $\mathcal{C}$. Fixed point theorems such as Brouwer's or Kakutani's, which are typically used in existence proofs for Nash equilibria, require the feasible set to be convex. In order to establish the existence of a fixed point for a best response map defined over our nonconvex $\mathcal{C}$, we rely on a

result that goes back to Eilenberg & Montgomery (1946), and Begle (1950), which relaxes the convexity requirement to the weaker topological notion of contractibility. A central part of our proof, Lemma 3.7, establishes that under Assumption 3.1, playerwise concavity of constraints implies contractibility of each connected component of $\mathcal{C}$.

Below, we define contractibility and state a version of Begle (1950)'s theorem applied to finite-dimensional Euclidean space. For examples of contractible sets, and a proof of Fact 3.6, see Appendix C.

**Definition 3.5.** Let $S \subseteq \mathbb{R}^n$. We say $S$ is contractible if there exists $s_0 \in S$, and a continuous map $H : [0, 1] \times S \to S$ such that for all $s \in S$, it holds that $H(0, s) = s$ and that $H(1, s) = s_0$.

**Fact 3.6.** *Let $X \subset \mathbb{R}^n$ be compact, contractible, and let $\phi : X \to X$ be continuous. Then there exists $x \in X$ such that $x = \phi(x)$, that is, $x$ is a fixed point of $\phi$.*

As seen in Example 3.3, the feasible region $\mathcal{C}$ may be disconnected; thus, it cannot be contractible. However, we establish the following key contractibility result.

**Lemma 3.7.** *Suppose Assumptions 2.2 and 3.1 hold. Let $\mathcal{C} \neq \emptyset$ and let $\mathcal{C}' \subseteq \mathcal{C}$ be any connected component of $\mathcal{C}$. Then $\mathcal{C}'$ is contractible.*

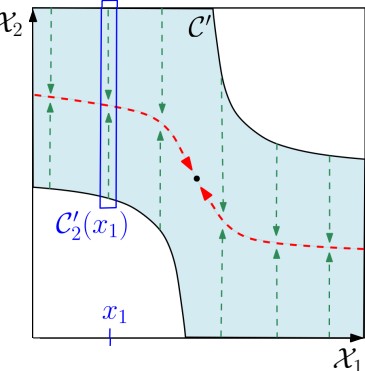

*Figure 2.* Proof sketch for contractibility (Lemma 3.7).

**Proof overview for Lemma 3.7** Because $\mathcal{C}'$ is not convex, we cannot simply contract it to a point via linear interpolation within $\mathcal{C}'$. However, we may leverage the playerwise concavity structure and construct a mapping $H : [0, 1] \times \mathcal{C}' \to \mathcal{C}'$ by sequentially contracting the feasible region along the dimensions of the players. For this overview, we focus on the two-player case which is illustrated in Figure 2. For $x_{-i} \in \mathcal{X}_{-i}$, let $\mathcal{C}'_i(x_{-i}) := \{x_i \in \mathcal{X}_i \mid (x_i, x_{-i}) \in \mathcal{C}'\}$ and $\text{supp}(\mathcal{C}'_i) := \{x_{-i} \in \mathcal{X}_{-i} \mid \mathcal{C}'_i(x_{-i}) \neq \emptyset\}$.

For $t \in [0, \frac{1}{2}]$ and $x = (x_1, x_2) \in \mathcal{C}'$, we let $t \mapsto H(t, x)$ linearly interpolate between $x$ and $(x_1, \bar{c}(\mathcal{C}'_2(x_1)))$ where for compact $S \subset \mathbb{R}^n$, $\bar{c}(S)$ denotes the centroid of $S$. This is depicted by the green arrows in Figure 2.

For $t \in [\frac{1}{2}, 1]$ and $x = (x_1, x_2) \in \mathcal{C}'$, we let $H(t, x)$ be the point $(x_1(t), \mathcal{C}'_2(x_1(t)))$ where $x_1(t)$ linearly interpolates between $x_1$ and $\bar{c}(\text{supp}(\mathcal{C}'_1))$. This is depicted by the red arrows in Figure 2.

In order to satisfy Definition 3.5, $H$ needs to be (a) well-defined, such that for all $t \in [0, 1]$ and $x \in \mathcal{C}'$, we have $H(t, x) \in \mathcal{C}'$, (b) continuous in $(t, x)$, and (c) must contract $\mathcal{C}'$ to some point $\hat{x} \in \mathcal{C}'$.

Property (a) is shown via convexity of the slices $\mathcal{C}'_i(x_{-i})$ which follows directly from the playerwise concavity of the constraints. Regarding (b), we observe that $\bar{c}(\cdot)$ and $\mathcal{C}'_i(\cdot)$, from which we compose $H$, are both continuous. In particular, continuity of the set-valued mapping $\mathcal{C}'_i(\cdot)$ follows from standard results in parametric optimization, which apply under the MFCQ condition (Assumption 3.1). For (c), by construction, for all $x \in \mathcal{C}'$, $H(0, x) = x$ and $H(1, x) = \bar{c}(\text{supp}(\mathcal{C}'_1))$.

Having established contractibility of $\mathcal{C}'$, we prove Theorem 3.4 via a fixed point argument.

**Proof overview for Theorem 3.4**   Our goal is to define a best response function $\Psi : \mathcal{C}' \to \mathcal{C}'$, to which Fact 3.6 can be applied. The usual construction of simultaneously mapping players' strategies to their best responses does not work here, as due to the nonconvexity of $\mathcal{C}'$, the resulting strategies may not be feasible. Instead, we again leverage the playerwise concave constraint structure by constructing $\Psi$ in a playerwise sequential manner. For each $i \in [m]$, suppose $u_i$ is strongly concave (see limit argument in the full proof, Appendix D, for why this is without loss of generality). Then $\Psi_i : \mathcal{C}' \to \mathcal{C}'$ with

$$\Psi_i(x) := \Big( \operatorname*{arg\,max}_{x'_i \in \mathcal{C}'_i(x_{-i})} u_i(x'_i, x_{-i}), \ x_{-i} \Big).$$

is well-defined since the maximizer is unique. Moreover, by a variant of Berge's maximum theorem for strongly concave objective functions over convex constraints (see Appendix C), $\Psi_i$ is continuous. We then apply Fact 3.6 to the composition $\Psi := \Psi_1 \circ \cdots \circ \Psi_m$. Using the playerwise concavity of constraints, it can be argued that any fixed point $x^\star \in \mathcal{C}'$ with $x^\star = \Psi(x^\star)$ is a constrained Nash equilibrium.

# 4. Learning Constrained Nash Equilibria

If players update their strategies using only local, gradient-based information, can they efficiently learn an approximate constrained Nash equilibrium? In this section, we investigate this question in potential games with shared coupling constraints, and propose methods for finding approximate constrained Nash equilibria in a deterministic, independent learning setting.

## 4.1. Constrained Potential Games

While Assumptions 2.2 and 3.1 suffice for existence, without additional assumptions on the game structure, the problem of finding $\epsilon$-approximate Nash equilibria is known to be PPAD-hard even for unconstrained normal-form games (Daskalakis et al., 2009; Rubinstein, 2015). Therefore, we focus on the subclass of potential games (Monderer & Shapley, 1996) with coupling constraints. For our purpose, it suffices to have a potential function defined over the feasible region.

**Definition 4.1.** A potential game with coupling constraints is a constrained game for which there exists a function $\Phi : \mathcal{C} \to \mathbb{R}$ such that for all players $i \in [m]$, strategy profiles $x \in \mathcal{C}$, and $x'_i \in \mathcal{C}_i(x_{-i})$, we have

$$\Phi(x) - \Phi(x'_i, x_{-i}) = u_i(x) - u_i(x'_i, x_{-i}).$$

Examples of such continuous potential games trivially include cooperative games, and, as shown in Example 4.2 below, mixed strategy extensions of finite action potential games with coupling constraints. This class of potential games also arises in economics, e.g. in the context of the Cournot competition (Cournot, 1838), in communication networks (Scutari et al., 2006), and in transportation (Altman & Wynter, 2004).

**Example 4.2.** Consider the game with joint action space $\mathcal{A} := \times_{i \in [m]} \mathcal{A}_i$ where each $\mathcal{A}_i$ is a finite set, utilities $\widetilde{u}_i : \mathcal{A} \to [0, 1]$ for $i \in [m]$, and constraints $\widetilde{c}_j : \mathcal{A} \to [0, 1]$ with thresholds $\alpha_j$, for $j \in [b]$, that induce a feasible region $\widetilde{\mathcal{C}} \subset \mathcal{A}$. Suppose there exists a potential function $\phi : \widetilde{\mathcal{C}} \to \mathbb{R}$ that satisfies $\phi(a) - \phi(a'_i, a_{-i}) = \widetilde{u}_i(a) - \widetilde{u}_i(a'_i, a_{-i})$ for all $i \in [m]$, $a \in \widetilde{\mathcal{C}}$, and $a'_i \in \widetilde{\mathcal{C}}_i(a_{-i})$. Then, the mixed strategy extension of this game with strategy space $\mathcal{X} := \times_{i \in [m]} \Delta(\mathcal{A}_i)$, utilities $u_i(x) := \mathbb{E}_{a \sim x}[\widetilde{u}_i(a)]$, and constraints $c_j(x) := \mathbb{E}_{a \sim x}[\widetilde{c}_j(a)]$ with thresholds $\alpha_j$ is a potential game with potential $\Phi(x) := \mathbb{E}_{a \sim x}[\phi(a)]$. See Appendix F for a proof of this fact.

## 4.2. Independent Learning Protocol

We assume players to interact with the game for $T$ rounds. In each round $t$, for $0 \le t \le T - 1$, player $i$ chooses some strategy $x_i^{(t)} \in \mathcal{X}_i$ and receives first-order feedback for the utility $\nabla_{x_i} u_i(x^{(t)})$, as well as zeroth- and first-order feedback for each constraint $j \in [b]$, that is, $c_j(x^{(t)})$ and $\nabla_{x_i} c_j(x^{(t)})$. Players do not observe other players' strategies or utility feedback. Moreover, besides the game interaction, no communication or coordination (e.g., shared randomness) among players is allowed.

## 4.3. Interior Point Method

Access to a feasible initial strategy is crucial, as otherwise even a two-player game with bilinear constraints requires

solving an NP-hard optimization problem (Witsenhausen, 1986). For iterate $x^{(t)} \in \mathcal{C}$, we denote its feasibility margin by $\beta^{(t)} := \min_{j \in [b]} \{c_j(x^{(t)}) - \alpha_j\}$.

**Assumption 4.3.** *The initial strategy $x^{(0)} \in \mathcal{X}$ is strictly feasible, i.e., $\beta^{(0)} > 0$, and for all $i \in [m]$, $x_i^{(0)}$ is initially known to player $i$.*

To ensure feasibility of the iterates during the algorithm, we use log barrier regularized gradient descent, an interior point approach recently studied for nonconvex problems with nonconvex constraints (Hinder & Ye, 2024; Usmanova et al., 2024). Under the assumed potential structure, we show that independently updated iterates remain in the interior of the joint feasible region.

**Log barrier regularization** First, we define the set $\mathcal{C}^{\circ} := \{x \in \mathcal{X} \mid \forall j \in [b], \ c_j(x) > \alpha_j\}$ of strictly feasible strategies. For each player $i \in [m]$ and $x \in \mathcal{C}^{\circ}$, let

$$B_i^{\eta}(x) := u_i(x) + \eta \sum_{j \in [b]} \log(c_j(x) - \alpha_j)$$

where $\eta > 0$ is a parameter controlling the regularization. Observe that for each constraint $j \in [b]$, the log term serves as a barrier that tends to $-\infty$ as $c_j(z)$ approaches the threshold $\alpha_j$. Then, we propose the following independent log barrier regularized gradient ascent: Suppose in iteration $t$ for $0 \le t \le T - 1$, each player $i$ simultaneously performs the projected regularized gradient update

$$x_i^{(t+1)} = \mathcal{P}_{\mathcal{X}_i} \left[ x^{(t)} + \gamma^{(t)} \nabla_{x_i} B_i^{\eta}(x) \right] \quad (1)$$

for some stepsize $\gamma^{(t)} > 0$, where $\nabla_{x_i} B_i^{\eta}(x) = \nabla_{x_i} u_i(x) + \eta \sum_{j \in [b]} \frac{\nabla_{x_i} c_j(x)}{c_j(x) - \alpha_j}$, and $\mathcal{P}_{\mathcal{X}_i}[\cdot]$ denotes projection onto $\mathcal{X}_i$. Our goal in the following is to design $\gamma^{(t)}$ and $\eta$ to guarantee feasibility of all joint iterates and convergence to a constrained Nash equilibrium.

### 4.4. Convergence Result

In order to prove our convergence result, we adopt standard assumptions on smoothness and Lipschitz continuity of utility functions, see also Mertikopoulos et al. (2024), and extend these regularity conditions to the constraints.

**Assumption 4.4.** *There exist constants $L$ and $M$ such that the functions $u_i$ for $i \in [m]$ and $c_j$ for $j \in [b]$ are all differentiable, $L$-Lipschitz continuous, and $M$-smooth in the joint strategies $x \in \mathcal{X}$.*

Despite smoothness of utilities and constraints, $B_i^{\eta}(x)$ is not globally smooth due to unbounded growth of the log term towards the boundary of $\mathcal{C}$, which represents a key challenge in the analysis. However, we show that $B_i^{\eta}(x)$ is smooth over the joint strategy trajectory of independent log barrier regularized gradient ascent when using suitable adaptive stepsizes $\gamma^{(t)}$.

**Choice of stepsizes** Stepsizes are chosen as

$$\gamma^{(t)} := \min \left\{ \min_{j \in [b]} \frac{c_j(x^{(t)}) - \alpha_j}{2mL^2}, \frac{1}{M_{\Phi}^{\eta, \beta^{(t)}/2}} \right\}, \quad (2)$$

where $M_{\Phi}^{\eta, \beta^{(t)}} := M\sqrt{m} + \eta M b (\beta^{(t)})^{-1} + \eta L^2 b (\beta^{(t)})^{-2}$. The first term in (2) ensures that iterates remain within the feasible region. The second term bounds $\gamma^{(t)}$ by the inverse of the local smoothness parameter $M_{\Phi}^{\eta, \beta^{(t)}}$. We further describe these terms in the proof overview of Theorem 4.7 below.

While for the existence result, it is sufficient to directly apply the standard MFCQ condition stated in Assumption 3.1, our convergence proof requires an extended version of this condition. Namely, a uniform constraint ascent direction must exist for all feasible joint strategies $x \in \mathcal{C}$ *near* the boundary of $\mathcal{C}$, ensuring that log barrier gradients consistently steer the iterates away from the boundary.

**Assumption 4.5.** *Let $\rho > 0$, and let $\mathcal{J}_{\rho}(x) := \{j \in [b] \mid c_j(x) - \alpha_j \le \rho\}$ be the set of $\rho$-approximately active constraints for strategy $x \in \mathcal{C}$. There exists $\ell > 0$, such that for all $x \in \mathcal{X}$ and all $j \in \mathcal{J}_{\rho}(x)$, there exists $\delta$ with $x + \delta \in \mathcal{X}$, $\|\delta\| \le 1$, such that $\langle \delta, \nabla c_j(x) \rangle > \ell$.*

**Remark** Note that for $\rho = 0$ and $\mathcal{X} = \mathbb{R}^n$, we recover the classic Mangasarian-Fromovitz constraint qualification (MFCQ) condition (Mangasarian & Fromovitz, 1967). Moreover, as shown by Ni & Kamgarpour (2025) (see their Appendix B.1), in our specific case of compact strategy spaces and Lipschitz continuous constraint functions, Assumption 3.1 in fact implies existence of *some* $\rho > 0$ and $\ell > 0$ for which Assumption 4.5 holds. We nevertheless state the extended version explicitly, since the convergence rate in Theorem 4.7 depends on $\rho$ and $\ell$, and the values implied by Assumption 3.1 may be arbitrarily small.

As the solution concept of our algorithms, we introduce an approximate notion of constrained Nash equilibrium. Note that the case $\epsilon = 0$ recovers the exact notion from Section 2.

**Definition 4.6.** *For $\epsilon > 0$, $x \in \mathcal{X}$ is a constrained $\epsilon$-approximate Nash equilibrium if $x \in \mathcal{C}$ and for all $i \in [m]$ and $x_i' \in \mathcal{C}_i(x_{-i})$, $u_i(x_i', x_{-i}) \le u_i(x) + \epsilon$.*

Now, we are ready to state our main theorem about convergence to a constrained Nash equilibrium.

**Theorem 4.7.** *Let $\epsilon > 0$ and set $\eta = \epsilon$. Let Assumptions 2.2, 4.3, 4.4, and 4.5 hold with $\rho \ge \eta$. For stepsizes $\{\gamma^{(t)}\}_{t=0}^{T-1}$ chosen as in (2), suppose we perform $T = \mathcal{O}(\epsilon^{-3})$ rounds of simultaneous updates as in (1). Then, there exists $0 \le t \le T$ such that $x^{(t)}$ is a constrained $\epsilon$-approximate Nash equilibrium.*

We first discuss how this result relates to the existing literature. Then we provide an overview of our main ideas and the challenges of our analysis; for the full proof we refer to Appendix F.

**Remarks**

1. The work of Anagnostides et al. (2022) also considers concave continuous action potential games, but without coupling constraints, establishing convergence to a Nash equilibrium at an $\mathcal{O}(\epsilon^{-2})$ best-iterate rate (see their Theorem B.6). Thus, handling coupling constraints via log barrier regularization comes at the cost of an additional $\mathcal{O}(\epsilon^{-1})$ factor. This is due to small adaptive stepsizes as iterates approach the boundary of the feasible region.

2. While we are not aware of any results on learning Nash equilibria in continuous action potential games with coupling constraints, a result related to ours exists for constrained Markov potential games under an independent learning protocol (Jordan et al., 2024). Unlike our approach, this result relies on a double-loop proximal-point-like algorithm that requires a so-called uniform Slater's condition. Intermediate iterates are not guaranteed to remain within the feasible region. In the exact-gradient setting, a best-iterate convergence to a constrained $\epsilon$-approximate Nash equilibrium at a slower $\mathcal{O}(\epsilon^{-4})$ rate is proven.

3. An appealing property of log barrier methods is the feasibility of all iterates, making them well-suited for problems where constraints model safety conditions that may not be violated during learning.

**Proof overview**   The main challenge of the analysis is to ensure feasibility of all iterates and smoothness of the objective along the trajectory despite the players' independent updates. Towards this end, we observe that due to the cooperative structure of the log barrier regularization, $\Phi^\eta(x) := \Phi(x) + \eta \sum_{j \in [b]} \log(c_j(x) - \alpha_j)$ is a potential function for the regularized game. Therefore, partial gradients are aligned with the potential gradient, i.e., $\nabla_{x_i} B_i^\eta(x) = \nabla_{x_i} \Phi^\eta(x)$ for all $i \in [m]$ which allows us to equivalently analyze our independent updates as a centralized log barrier regularized gradient ascent on $\Phi^\eta$.

Next, we reason about feasibility of the joint strategy iterates and smoothness of $\Phi^\eta$ along the gradient ascent trajectory. For $\beta^{(t)}$-feasible $x^{(t)}$, we give a Hessian bound $\|\nabla^2 \Phi^\eta(x)\| \leq M_\Phi^{\eta, \beta^{(t)}}$. Moreover, we show that if an iterate $x^{(t)}$ is $\beta^{(t)}$-feasible, and stepsizes are chosen as in (2), then all strategies on the line segment $[x^{(t)}, x^{(t+1)}]$ are $(\beta^{(t)}/2)$-feasible. Consequently, $\Phi^\eta$ is $(M_\Phi^{\eta, \beta^{(t)}/2})$-smooth over $[x^{(t)}, x^{(t+1)}]$. Therefore, if we additionally

ensure $\gamma^{(t)} \leq 1/(M_\Phi^{\eta, \beta^{(t)}/2})$, as in (2), a standard inequality from nonconvex smooth optimization (Bubeck et al. (2015), Lemma 3.6) shows sufficient increase proportional to $\gamma^{(t)}$. Nevertheless, $M_\Phi^{\eta, \beta^{(t)}/2}$ may grow arbitrarily and lead to diminishing stepsizes. Leveraging Assumption 4.5, we give a lower bound on $\gamma^{(t)}$ based on the fact that the log barrier term dominates and steers iterates away from the boundary of $\mathcal{C}$ whenever the feasibility margin $\beta^{(t)}$ is too small.

As a result, for our choices of $\eta$ and $T$, we obtain best-iterate convergence at an $\mathcal{O}(\epsilon^{-3})$ rate towards an $\epsilon$-stationary point $x^\star$ of $\Phi^\eta$. This implies a set of $\epsilon$-approximate KKT conditions to hold for the constrained problem $\max_{x \in \mathcal{C}} \Phi^\eta(x)$ at $x^\star$, which means $x^\star$ is an $\epsilon$-approximate constrained Nash equilibrium.

## 5. Simulations

To demonstrate the convergence of our method, we conduct numerical simulations on two continuous potential games with shared coupling constraints.[2] The first is a simple two-player cooperative game played over a nonconvex feasible region. The second is a routing problem with link capacity constraints. We measure convergence in terms of Nash-Gap$_i(x) := \max_{x_i' \in \mathcal{C}_i(x_{-i})} u_i(x_i', x_{-i}) - u_i(x)$ for each player $i$. This can be computed by solving the single-player constrained optimization problem where $x_{-i}$ remains fixed. We define Nash-Gap$(x) := \max_{i \in [m]}$ Nash-Gap$_i(x)$. This serves as a measure of convergence, as for any $\epsilon > 0$, if Nash-Gap$(x) := \max_{i \in [m]}$ Nash-Gap$_i(x) \leq \epsilon$, then $x$ is a constrained $\epsilon$-approximate Nash equilibrium.

**Cooperative game**   Consider a two-player game where the strategy $x \in \mathcal{X} = [0, 1]^2$ models each player's contribution to a cooperative task for which utilities are given by

$$u_1(x_1, x_2) = u_2(x_1, x_2)$$
$$= \underbrace{(x_1 + x_2)}_{(a)} - a \underbrace{(x_1 - x_2)^2}_{(b)} - b \underbrace{(x_1^2 + x_2^2)}_{(c)}$$

for some $a, b > 0$. One may interpret term (a) as capturing the mutual benefit of the joint effort, term (b) as encouraging equal contribution, and term (c) as reflecting diminishing returns of individual high efforts. Additionally, we introduce a shared constraint by defining $c(x_1, x_2) := 1 - x_1 x_2$ and threshold $\alpha = 0.85$, representing a bound on the joint contribution.

Starting from an initial feasible point, we apply our log barrier regularized independent gradient ascent to the constrained potential game. Figure 3 visualizes the joint strategy trajectory within the utility landscape where colors indicate the log barrier value. The trajectory (in red) remains

---

[2]The code for our simulations is available at https://github.com/philip-jordan/log-barrier-cNE.

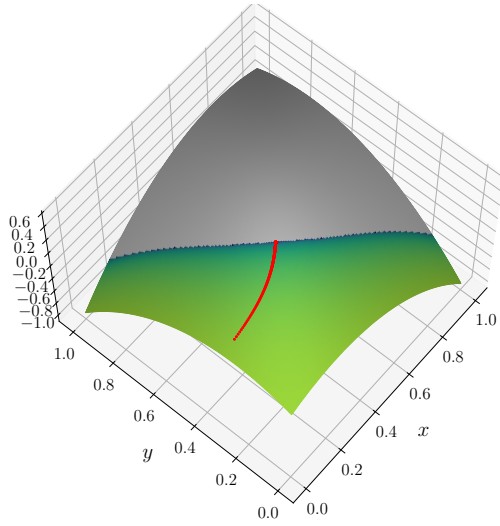

Figure 3. Strategy trajectory in the utility landscape of the cooperative game; colors indicate the log barrier value; grey is infeasible.

within the nonconvex feasible region throughout the iterations, aligning with our theoretical observations. In Figure 4, we plot the Nash-Gap$_i(x)$ for each player $i \in [1, 2]$ and observe convergence to a strategy near the boundary of $\mathcal{C}$ that forms an approximate constrained Nash equilibrium.

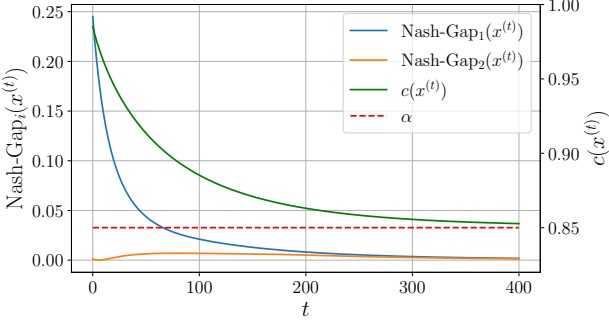

Figure 4. Convergence of log barrier regularized gradient ascent in the cooperative game: individual Nash gaps Nash-Gap$_i(x^{(t)})$ decrease to zero while the constraint $c(x^{(t)})$ remains below the threshold $\alpha$, indicating feasibility throughout.

**Network routing game** Beyond the cooperative case, we turn to a classical distributed routing problem, a canonical example of a potential game well studied in the literature on network routing (Menache & Ozdaglar, 2011). We extend this model by incorporating coupled capacity constraints. Specifically, we adopt an example of Tatarenko (2017); see their Example 2.2. Five players, one corresponding to each color in Figure 5, aim to route some amount of flow through five paths in the network, each associated to the respective player ($R_1$: blue, $R_2$: red, $R_3$: brown, $R_4$: orange, $R_5$: green). For $i \in [5]$, we let $x_i \in [0, 10]$ denote the flow on player $i$'s path $R_i$.

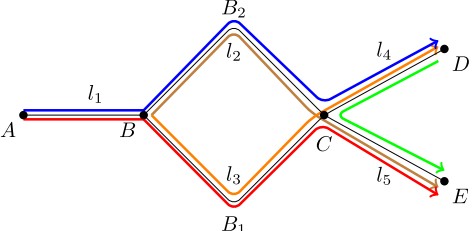

Figure 5. Routing game (Tatarenko (2017), Ex. 2.2).

Utilities represent a players' (linear) profit minus link costs, where the latter depends on the congestion of the used links. Concretely,

$$u_i(x) := px - \sum_{k:l_k \in R_i} P_k \left( \sum_{j:l_k \in R_j} x_j \right)$$

where we set $p = 42$ and $P_1(x) = 5x^2 + x$, $P_2(x) = 4x^2 + 2x + 1$, $P_3(x) = 5x^2$, $P_4(x) = 2x^2 + x$, $P_5(x) = 10x$. We additionally define shared capacity constraints on the links $l_2$ and $l_5$, namely

$$c_1(x) := \sum_{j:l_2 \in R_j} x_j, \qquad c_2(x) := \sum_{j:l_5 \in R_j} x_j$$

with upper bounds $\alpha_1 = 2.7$ and $\alpha_2 = 7$. Note that by negating, these constraints can be formulated as lower bounds in order to match our problem definition. Moreover, utilities and constraints are playerwise concave due to linearity and convexity of the functions $P_l$ for $l \in [5]$, hence satisfying Assumption 2.2.

Starting from feasible flows $x_i = \frac{1}{2}$ for all $i \in [5]$, Figure 6 shows the convergence of log barrier regularized independent gradient ascent to a constrained Nash equilibrium of the routing game. We point out that, again, all iterates remain within the feasible region.

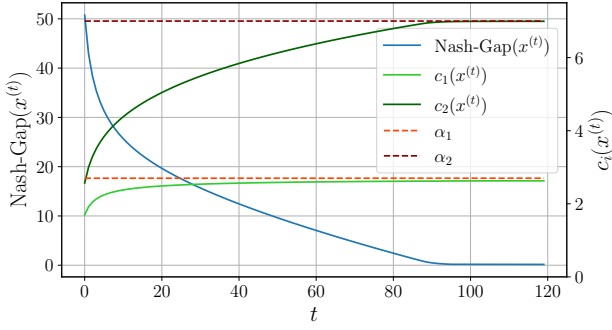

Figure 6. Convergence in the constrained routing game: the Nash gap decreases to zero while link capacity constraints $c_1$ and $c_2$ remain below their respective thresholds $\alpha_1$ and $\alpha_2$ throughout all iterations.

# 6. Conclusion

In this paper, we established existence of Nash equilibria in continuous action games with playerwise concave utilities and shared, playerwise concave constraints. Furthermore, we proposed an independent log barrier regularized gradient method and proved its convergence to constrained Nash equilibria in potential games with shared coupling constraints.

Our findings suggest several directions for future work. First, characterizing equilibrium existence and computation in settings where utilities or constraints lack playerwise concavity, such as Markov games, remains to be further explored. Second, extending our algorithmic approach to accommodate stochastic zeroth- or first-order feedback is a natural next step. Third, adapting our framework to learn equilibria in other tractable subclasses of playerwise concave games, including zero-sum and monotone games, is another direction worth exploring.

# Acknowledgements

This work was supported in part by the Swiss National Science Foundation under Grant 207984.

# Impact Statement

This paper presents work whose goal is to advance the field of Machine Learning. There are many potential societal consequences of our work, none which we feel must be specifically highlighted here.

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

# A. Overview of Notation

Table 1 gives an overview of our notation. All notations are introduced before their first use as well.

*Table 1.* Overview of notation

| | | |
|---|---|---|
| $\Delta(X)$ | $:=$ | probability simplex over finite set $X$ |
| $[n]$ | $:=$ | the set $\{1, 2, \ldots, n\}$ for $n \in \mathbb{N}$ |
| $S^\circ$ | $:=$ | interior of the set $S \subset \mathbb{R}^n$ |
| $\mathcal{P}_S[x]$ | $:=$ | projection of $x \in \mathbb{R}^n$ onto compact convex $S \subset \mathbb{R}^n$ |
| $[y]_+$ | $:=$ | $\max\{0, y\}$ for $y \in \mathbb{R}$ |
| $\|x\|$ | $:=$ | $\ell_2$-norm of $x \in \mathbb{R}^n$ |
| $[x, x']$ | $:=$ | $\{\lambda x + (1 - \lambda)x' \mid \lambda \in [0, 1]\} \subset \mathbb{R}^n$ for $x, x' \in \mathbb{R}^n$ |
| $\mathrm{dist}(x, S)$ | $:=$ | $\inf_{x' \in S} \|x - x'\|$ for $S \subset \mathbb{R}^n$ and $x \in S$ |
| $\mathrm{conv}(S)$ | $:=$ | convex hull of a set $S \subset \mathbb{R}^n$ |
| $\mathcal{X}$ | $:=$ | $\times_{i \in [m]} \mathcal{X}_i$, set of strategy profiles |
| $u_i(x)$ | $:=$ | player $i$'s utility function $u_i : \mathcal{X} \to [0, 1]$ |
| $c_j(x)$ | $:=$ | $j$-th shared cost function $c_j : \mathcal{X} \to [0, 1]$ |
| $\mathcal{C}$ | $:=$ | set of feasible strategy profiles, $\mathcal{C} \subset \mathcal{X}$ |
| $\mathcal{C}_i(x_{-i})$ | $:=$ | $\{x_i \in \mathcal{X}_i \mid (x_i, x_{-i}) \in \mathcal{C}\}$ for $x_{-i} \in \mathcal{X}_{-i}$ |
| $\mathrm{supp}(\mathcal{C}_i)$ | $:=$ | $\{x_{-i} \in \mathcal{X}_{-i} \mid \exists x_i \in \mathcal{X}_i \text{ s.t. } (x_i, x_{-i}) \in \mathcal{C}\}$ |
| $\beta^{(t)}$ | $:=$ | $\min_{j \in [b]} \{c_j(x^{(t)}) - \alpha_j\}$ |
| $\gamma^{(t)}$ | $:=$ | stepsize in iteration $t$ for (non-optimistic) log barrier gradient ascent |
| $B_i^\eta(x)$ | $:=$ | $u_i(x) + \eta \sum_{j \in [b]} \log(c_j(x) - \alpha_j)$ |
| $\Phi(x)$ | $:=$ | potential function for $m$-player game in Section 4 |
| $\Phi^\eta(x)$ | $:=$ | $\Phi(x) + \eta \sum_{j \in [b]} \log(c_j(x) - \alpha_j)$; potential of regularized game |
| $M_\Phi^{\eta, \beta}$ | $:=$ | Hessian norm bound for $\Phi^\eta(x)$ for $\beta$-feasible $x \in \mathcal{X}$ |

# B. Discussion on Constraint Structure

In this section, we provide further motivation for focusing our paper on the case of shared coupling constraints. We first define the setting with non-shared constraints. Then, we provide an example of a game for which no constrained Nash equilibrium exists.

**Game setting**  Consider the continuous static game setting with players $[m]$, joint strategy space $\mathcal{X} = \times_{i \in [m]} \mathcal{X}_i$, and utilities $u_i(x) : \mathcal{X} \to \mathbb{R}$ for each $i \in [m]$, as before. Non-shared coupling constraints are introduced through the set of functions and thresholds $\{(c_{i,j}, \alpha_{i,j})\}_{i \in [m], j \in [b_i]}$ where for each $i \in [m]$, $b_i$ is the number of constraints of player $i$. Moreover, for all $i \in [m]$ and $j \in [b_i]$, we have $\alpha_{i,j} \in \mathbb{R}$ and $c_{i,j} : \mathcal{X} \to \mathbb{R}$ continuous. The set of feasible strategy profiles is then given by

$$\widetilde{\mathcal{C}} := \left\{ x \in \mathcal{X} \mid \forall i \in [m], j \in [b_i], c_{i,j}(x) \geq \alpha_{i,j} \right\}.$$

For $i \in [m]$ and $x_{-i} \in \mathcal{X}_{-i}$, let the set of feasible responses of player $i$ to $x_{-i}$ be denoted by $\widetilde{\mathcal{C}}_i(x_{-i}) := \{x_i \in \mathcal{X}_i \mid \forall j \in [b_i], c_{i,j}(x) \geq \alpha_{i,j}\}$.

**Nash equilibria**  A constrained Nash equilibrium in the non-shared constraint setting is a strategy profile $x \in \mathcal{X}$ such that $x \in \widetilde{\mathcal{C}}$, and for all $i \in [m]$ and playerwise deviations $x_i' \in \widetilde{\mathcal{C}}_i(x_{-i})$, it holds that $u_i(x_i', x_{-i}) \leq u_i(x)$.

We point out that in above definition, deviations of player $i$ only need to be feasible with respect to player $i$'s constraints; this is what distinguishes non-shared constraints from shared ones. Note that this equilibrium notion matches the one introduced by Debreu (1952). The nonemptiness assumption of Debreu (1952) then requires that for all $i \in [m]$ and $x_{-i} \in \mathcal{X}_{-i}$, we have $\mathcal{C}_i(x_{-i}) \neq \emptyset$. Under this assumption, Debreu (1952) proves existence of constrained Nash equilibria.

Is it possible to extend Debreu (1952)'s theorem and establish existence results based on playerwise concavity (our Assumption 2.2) in this class of games with non-shared constraints, without introducing further assumptions such as the above nonemptiness condition on the feasible response sets? We give a negative answer in the form of the following counterexample. Note that the counterexample even has jointly linear constraints and utilities.

**Proposition B.1.** *There exists a two-player constrained game with non-shared playerwise concave constraints $c_{1,1}$ and $c_{2,1}$ that has no constrained Nash equilibrium.*

*Proof.* Let $\mathcal{X}_1 = \mathcal{X}_2 = [0, 1]$ and define utilities and costs as follows:

$$\begin{aligned} u_1(x_1, x_2) &= -x_1, \\ u_2(x_1, x_2) &= -x_2, \\ c_{1,1}(x_1, x_2) &= x_2, \\ c_{2,1}(x_1, x_2) &= x_1, \end{aligned}$$

with thresholds $\alpha_{1,1} = \alpha_{1,2} = 0.5$. Assumption 2.2 on the playerwise concavity of utilities and constraints holds due to linearity of utility and constraint functions. The joint feasible region is

$$\widetilde{\mathcal{C}} = \left\{ x \in [0, 1]^2 \mid x_1 \geq 0.5 \wedge x_2 \geq 0.5 \right\},$$

and for any $(x_1, x_2) \in [0, 1]^2$, the feasible response sets are given by $\widetilde{\mathcal{C}}_2(x_1) = \widetilde{\mathcal{C}}_1(x_2) = [0.5, 1]$. Observe that for any $x \in \widetilde{\mathcal{C}}$, player 1 can increase its utility by deviating to $x_1' = 0 \in \widetilde{\mathcal{C}}_1(x_2)$. Hence no constrained Nash equilibrium can exist. $\qquad\square$

Note that Debreu (1952)'s assumption prevents constructing such pathological cases: In above example, the nonemptiness condition is violated, as for any $\hat{x} \in [0, 0.5)^2$, $\widetilde{\mathcal{C}}_1(\hat{x}_2) = \widetilde{\mathcal{C}}_2(\hat{x}_1) = \emptyset$. However, as seen in our Example 3.3, Debreu (1952)'s nonemptiness assumption is strong and captures only a limited class of coupling constraints.

As we aim to avoid making further assumptions besides playerwise concavity, the above proposition motivates focusing on the case of shared constraints in this paper.

# C. Background for Section 3

In this section, we provide the background required for presenting a formal proof of our main existence result, Theorem 3.4.

## C.1. Contractibility

First, we restate the definition of contractibility, and provide examples to illustrate the concept.

**Definition 3.5.** Let $S \subseteq \mathbb{R}^n$. We say $S$ is contractible if there exists $s_0 \in S$, and a continuous map $H : [0, 1] \times S \to S$ such that for all $s \in S$, it holds that $H(0, s) = s$ and that $H(1, s) = s_0$.

For example, observe that every nonempty convex set $X \subseteq \mathbb{R}^n$ is contractible, by choosing arbitrary $x_0 \in X$ and setting $H(t, x) = (1 - t)x + tx_0$. By convexity, for all $t \in [0, 1]$ and $x \in X$, it holds that $H(t, x) \in X$. Star-shaped sets are contractible by the same argument when choosing $x_0$ as the star center.

## C.2. Continuity for Set-Valued Functions

We next recall continuity concepts for set-valued functions, which will be used when discussing continuity properties of feasible response mappings.

**Definition C.1.** Let $Y, Z \subseteq \mathbb{R}^n$, and let $F : Y \to \mathcal{P}(Z)$ be a set-valued function.

1. The function $F$ is *upper semicontinuous* at a point $y_0 \in Y$ if and only if for any open subset $V \subseteq Z$ with $F(y_0) \subseteq V$ there exists a neighborhood $U(y_0)$ of $y_0$, such that $F(y) \subseteq V$ for all $y \in U(y_0)$.

2. The function $F$ is *lower semicontinuous* at a point $y_0 \in Y$ if and only if for any open subset $V \subseteq Z$ with $F(y_0) \cap V \neq \emptyset$ there exists a neighborhood $U(y_0)$ of $y_0$, such that $F(y) \cap V \neq \emptyset$ for all $y \in U(y_0)$.

3. The function $F$ is continuous if and only if it is both upper and lower semicontinuous.

To introduce a notion of Lipschitz continuity for set-valued functions, we first define the *Hausdorff distance* between two nonempty sets $Y, Z \subseteq \mathbb{R}^n$ as

$$d_H(Y, Z) := \max \left\{ \sup_{y \in Y} \operatorname{dist}(y, Z), \ \sup_{z \in Z} \operatorname{dist}(z, Y) \right\}$$

where $\operatorname{dist}(y, Z) := \inf_{z \in Z} \|y - z\|$.

**Definition C.2.** Let $Y, Z \subseteq \mathbb{R}^n$ and $L \geq 0$. The set-valued function $F : Y \to \mathcal{P}(Z)$ is said to be *L-Lipschitz continuous with respect to the Hausdorff distance* if for all $y_1, y_2 \in Y$, we have

$$d_H(F(y_1), F(y_2)) \leq L \|y_1 - y_2\|.$$

## C.3. Berge's Maximum Theorem

Next, we provide a statement of Berge's maximum theorem that is used in our existence proof. This particular variant considers maximization of *strongly* concave parameterized functions over a parameterized constraint set. For reference, see for instance the version stated in Sundaram (1996), Theorem 9.17.

**Theorem C.3.** *Let $X, \Theta \subseteq \mathbb{R}^n$, $f : X \times \Theta \to \mathbb{R}$, and $C : \Theta \to \mathcal{P}(X)$ be continuous, and for all $\theta \in \Theta$, let $C(\theta)$ be compact and nonempty. Suppose also that for all $\theta \in \Theta$, $f(\cdot, \theta)$ is strongly concave, and that $C(\theta)$ is convex. Then*

1. *$f^\star : \Theta \to \mathbb{R}$ with $f^\star(\theta) := \max_{x \in C(\theta)} f(x, \theta)$ is well-defined and continuous, and*

2. *the function $C^\star : \Theta \to \mathcal{P}(X)$ defined by*

$$C^\star(\theta) := \arg \max_{x \in C(\theta)} f(x, \theta)$$
$$= \{x \in C(\theta) \mid f(x, \theta) = f^\star(\theta)\}$$

*is single-valued and continuous.*

### C.4. Fixed Point Theorem

We conclude this section by providing a proof of Fact 3.6, the fixed point theorem used in our existence proof. It follows by applying a more general statement, namely Begle's fixed point theorem (Begle, 1950), to our particular case of finite-dimensional Euclidean space.

**Fact 3.6.** *Let $X \subset \mathbb{R}^n$ be compact, contractible, and let $\phi : X \to X$ be continuous. Then there exists $x \in X$ such that $x = \phi(x)$, that is, $x$ is a fixed point of $\phi$.*

*Proof.* As pointed out by Debreu (1952), in finite dimension, Begle's fixed point theorem applies to compact metric spaces that are both contractible and locally contractible. Local contractibility is a property of the space's local geometry that holds for any subset of Euclidean space (see notes by Wilkins (2017)). Hence, it suffices to assume contractibility. The statement then follows from Begle (1950), Theorem 1. Note that we state the fact for single-valued functions which directly follows from the set-valued version. □

## D. Proof of Existence Result

In this section, we provide the formal proof of Theorem 3.4, based on the background introduced in Appendix C. First, we show playerwise convexity of the feasible slices.

**Lemma D.1.** *Suppose Assumption 2.2 holds. Let $\mathcal{C} \neq \emptyset$, and let $\mathcal{C}' \subseteq \mathcal{C}$ be a connected component of $\mathcal{C}$. For any $i \in [m]$ and $x_{-i} \in \mathcal{X}_{-i}$, the set*

$$\mathcal{C}'_i(x_{-i}) \coloneqq \{x_i \in \mathcal{X}_i \mid (x_i, x_{-i}) \in \mathcal{C}'\}$$

*is convex.*

*Proof.* If $\mathcal{C}'_i(x_{-i}) = \emptyset$, then convexity trivially holds. Otherwise, let $x_i, x'_i \in \mathcal{C}'_i(x_{-i})$ be arbitrary. We know that $c_j(x_i, x_{-i}) \geq \alpha_j$ and $c_j(x'_i, x_{-i}) \geq \alpha_j$ for all $j \in [b]$. By concavity of each $c_j$, we also have $c_j(x_i^\lambda, x_{-i}) \geq \alpha_j$ for any $x_i^\lambda \coloneqq \lambda x_i + (1 - \lambda)x'_i$ with $\lambda \in [0, 1]$. Hence $(x_i^\lambda, x_{-i}) \in \mathcal{C}$. Moreover, $(x_i^\lambda, x_{-i})$ must be in the same connected component of $\mathcal{C}$ as $(x_i, x_{-i})$, otherwise we get a contradiction with the fact that the line segment $[(x_i, x_{-i}), (x_i^\lambda, x_{-i})] \subset \mathcal{C}$. Therefore, we conclude that $(x_i^\lambda, x_{-i}) \in \mathcal{C}'$ which proves convexity of $\mathcal{C}'_i(x_{-i})$. □

Let $\operatorname{supp}(\mathcal{C}'_i) \coloneqq \{x_{-i} \in \mathcal{X}_{-i} \mid \exists x_i \in \mathcal{X}_i \text{ s.t. } (x_i, x_{-i}) \in \mathcal{C}'\}$. We show continuity of $\mathcal{C}'_i$ over this domain as follows.

**Lemma D.2.** *Suppose Assumption 3.1 holds. Let $\mathcal{C} \neq \emptyset$, and let $\mathcal{C}' \subseteq \mathcal{C}$ be a connected component of $\mathcal{C}$. Then, for all $i \in [m]$, the set-valued function $\mathcal{C}'_i(\cdot)$ is continuous (i.e. both upper and lower semicontinuous) over $\operatorname{supp}(\mathcal{C}'_i)$.*

*Proof.* We apply Lemma 6.1 of Still (2018)'s notes on parametric optimization. In our setting, it states that if the MFCQ condition holds at every $x \in \mathcal{C}'$ (which is given by our Assumption 3.1), then the parametric set-valued function $\mathcal{C}'_i(\cdot)$ is $L$-Lipschitz continuous in terms of the Hausdorff distance over $\operatorname{supp}(\mathcal{C}'_i)$ for large enough $L > 0$. It is well-known that for closed-valued functions such Hausdorff continuity implies upper and lower semicontinuity (see e.g. Clarke et al. (1998), 7.29). □

Next, by using Lemma D.1 and Lemma D.2, we give a proof of the following contractibility lemma introduced in Section 3, which we restate here.

**Lemma 3.7.** *Suppose Assumptions 2.2 and 3.1 hold. Let $\mathcal{C} \neq \emptyset$ and let $\mathcal{C}' \subseteq \mathcal{C}$ be any connected component of $\mathcal{C}$. Then $\mathcal{C}'$ is contractible.*

*Proof.* For a compact set $X \subset \mathbb{R}^n$, its centroid is defined as

$$\bar{c}(X) \coloneqq \frac{1}{\operatorname{vol}(X)} \int_X x \, dx$$

where $\operatorname{vol}(X)$ is the $n$-dimensional volume of $X$ which is assumed to be positive[3].

---

[3] Note that if $X$ collapses to a dimension lower than $n$, the centroid remains well-defined by using the volume of the respective dimension. If $X$ is a single point, then $\bar{c}(X)$ is that point.

Towards constructing a contraction map, for $i \in [m]$, let $h_i : [0,1] \times \mathcal{C}' \to \mathcal{C}'$ be defined as

$$h_i(t,x) := (1-t)x + t\left(\bar{c}(\mathcal{C}'_i(x_{-i})), x_{-i}\right),$$

which can be seen as a mapping that contracts $\mathcal{C}'$ along player $i$'s dimension. Next, we want to sequentially combine these functions in a way that for $t \in [\frac{i-1}{m}, \frac{i}{m}]$, the playerwise contraction is applied according to $h_i$.

Let $\varphi_i(t) : [\frac{i-1}{m}, \frac{i}{m}] \to [0,1]$ be defined as $\varphi_i(t) := mt - i + 1$ which will be used as an auxiliary rescaling function. Its inverse is well-defined and will be denoted by $\varphi_i^{-1}(t) : [0,1] \to [\frac{i-1}{m}, \frac{i}{m}]$. Then, for $i \in [m]$, we define $H_i : (\frac{i-1}{m}, \frac{i}{m}] \times \mathcal{C}' \to \mathcal{C}'$ as

$$H_1(t,x) := h_1(\varphi_1(t), x)$$
$$H_i(t,x) := h_i\left(\varphi_i(t), H_{i-1}\left(\varphi_{i-1}^{-1}(1), x\right)\right)$$

Finally, our contraction map $H : [0,1] \times \mathcal{C}' \to \mathcal{C}'$ is defined as

$$H(t,x) := \begin{cases} x, & \text{if } t = 0; \\ H_i(t,x), & \text{if } t \in (\frac{i-1}{m}, \frac{i}{m}] \text{ for } i \in [m]. \end{cases}$$

To prove that $H$ is indeed a contraction, it remains to show:

(a) **Well-definedness:** For all $t \in [0,1], x \in \mathcal{C}', H(t,x) \in \mathcal{C}'$.

(b) **Continuity:** $H$ is continuous in $(t,x)$.

(c) **Contraction to point:** There exists $\hat{x} \in \mathcal{C}'$ such that for all $x \in \mathcal{C}', H(0,x) = x$ and $H(1,x) = \hat{x}$.

Regarding (a), observe that each $h_i$ is well-defined: By definition, we have $x_i \in \mathcal{C}'_i(x_{-i})$. By convexity of $\mathcal{C}'_i(x_{-i})$, as shown in Lemma D.1, it holds that $\bar{c}(\mathcal{C}'_i(x_{-i})) \in \mathcal{C}'_i(x_{-i})$. Thus for all $t \in [0,1]$, again by convexity, $(1-t)x_i + t\,\bar{c}(\mathcal{C}'_i(x_{-i})) \in \mathcal{C}'$. As $h_i$ keeps $x_{-i}$ fixed, we have $h_i(t,x) \in \mathcal{C}'$ for all $t \in [0,1]$ and $x \in \mathcal{C}'$. Then, by definition, $H_1$ is well-defined, and the same follows for each $H_i$ inductively. We conclude that $H$ is well-defined, as it is piecewise equivalent to the $H_i$'s.

To show (b), note that by Lemma D.2, $\mathcal{C}'_i(x_{-i})$ is continuous in $x_{-i}$. Then, continuity of each $h_i$ in $(t,x)$ follows together with the continuity of the centroid function $\bar{c}(\cdot)$. Similarly as in (a), the recursive definitions of each $H_i$ and the piecewise definition of $H$ lets us conclude that $H$ is indeed continuous in $(t,x)$. In particular, we point out that continuity at the boundary of the pieces is ensured, as for all $i \in [m] \setminus \{1\}$ and all $x \in \mathcal{C}'$, by definition, $\lim_{t \to (i-1)/m^+} H_i(t,x) = H_{i-1}(\varphi_{i-1}^{-1}(1), x)$.

For (c), by definition we have $H(0,x) = x$ for all $x \in \mathcal{C}$. To argue about $H(1,\cdot)$, let $\mathcal{C}'^{(0)} := \mathcal{C}'$ and, for $i \in [m]$, let $\mathcal{C}'^{(i)} \subseteq \mathcal{C}'$ denote the image of the function $x \mapsto H_i(1,x)$. Thus, $\mathcal{C}'^{(i)} := \{H_i(\varphi_i^{-1}(1), x) \mid x \in \mathcal{C}'\}$ which due to the recursive definition of $H_i$, for all $i \in [m]$, is equal to $\{h_i(1,x) \mid x \in \mathcal{C}'^{(i-1)}\}$. Let $d_i := \dim(\mathcal{X}_i)$ and notice that $\dim(\mathcal{C}'^{(0)}) \leq \sum_{i \in [m]} d_i$. Next, we have $\mathcal{C}'^{(1)} = \{(\bar{c}(\mathcal{C}'_1(x_{-i})), x_{-i}) \mid \exists x_1 \in \mathcal{X}_1 \text{ s.t. } (x_1, x_{-1}) \in \mathcal{C}'^{(0)}\}$, meaning $\mathcal{C}'^{(1)}$ is can be described as a parameterized set with parameters from a set of dimension at most $\sum_{i=2}^m d_i$. Hence $\dim(\mathcal{C}'^{(1)}) \leq \sum_{i=2}^m d_i$. Similarly, we can argue that for all $1 < i \leq [m]$, $\dim(\mathcal{C}'^{(i)}) \leq \sum_{j=i+1}^m d_i$, since $\mathcal{C}'^{(i)}$ can be described as a parameterized set where coordinates corresponding to the first $i$ players are determined as a function of the remaining players $i+1$ to $m$. In particular, for the image of $x \mapsto H(1,x)$, namely the set $\mathcal{C}'^{(m)}$, we have therefore have $\dim(\mathcal{C}'^{(m)}) \leq 0$. Finally, in order to argue that $\mathcal{C}'^{(m)}$ is connected, we recall that the image of any continuous function with connected domain is connected. Since $\mathcal{C}'$ was chosen as a connected component, by continuity of $x \mapsto H(1,x)$, $\mathcal{C}'^{(m)}$ is therefore connected. As a connected set of dimension 0, $\mathcal{C}'^{(m)}$ can only contain a single point $\hat{x}$, meaning $H(1,x) = \hat{x}$ for all $x \in \mathcal{C}$. □

We are now ready to prove our main result on existence of constrained Nash equilibria.

**Theorem 3.4.** *Let Assumptions 2.2 and 3.1 hold, let $\mathcal{C} \neq \emptyset$. Then there exists a constrained Nash equilibrium $x^\star \in \mathcal{X}$.*

*Proof.* Let $\mathcal{C}' \subseteq \mathcal{C}$ be any connected component of $\mathcal{C}$, which exists as $\mathcal{C} \neq \emptyset$.

For any $\epsilon > 0$, define the perturbed game $\Gamma^\epsilon$ with regularized utilities $u_i^\epsilon(x) := u_i(x) - \epsilon \|x_i\|^2$; constraints are left unchanged. Observe that $\Gamma^\epsilon$ has playerwise strongly concave utilities. We first show that for any $\epsilon > 0$, $\Gamma^\epsilon$ has a constrained Nash equilibrium in $\mathcal{C}'$. Then we obtain the existence result for the original game via a limit argument.

Let $\epsilon > 0$. For each $i \in [m]$, define the playerwise best response map $\Psi_i : \mathcal{C}' \to \mathcal{C}'$ as

$$\Psi_i(x) := \left( \underset{x_i' \in \mathcal{C}_i'(x_{-i})}{\arg\max} \ u_i^\epsilon(x_i', x_{-i}), \ x_{-i} \right).$$

By a variant of Berge's maximum theorem for strongly concave objective functions over convex constraints (see Theorem C.3 in Appendix C), $\Psi_i$ is continuous and single-valued. Define the joint best response map $\Psi : \mathcal{C}' \to \mathcal{C}'$ as the composition $\Psi := \Psi_1 \circ \cdots \circ \Psi_m$, which is also continuous.

Since $\mathcal{C}'$ is compact and contractible (by Lemma 3.7), it follows from Fact 3.6 that there exists $x^\star \in \mathcal{C}'$ such that $x^\star = \Psi(x^\star)$. As each $\Psi_i$ leaves the strategies of players other than $i$ unchanged, this implies that $\Psi_i(x^\star) = x^\star$ for all $i \in [m]$. Suppose $x^\star$ is not a constrained Nash equilibrium. Then for some $i \in [m]$, there exists $x_i'$ such that $\hat{x} := (x_i', x_{-i}^\star) \in \mathcal{C}$ and $u_i(\hat{x}) > u_i(x^\star)$. We observe that by playerwise concavity of constraints, the line segment $[x^\star, \hat{x}] \subset \mathcal{C}$ satisfies all constraints, meaning $x^\star$ and $\hat{x}$ lie in the same connected component of $\mathcal{C}$; thus $\hat{x} \in \mathcal{C}' \subseteq \mathcal{C}$. We have reached a contradiction with $x_i^\star$ being the maximizer in the definition of $\Psi_i(x^\star)$, hence $x^\star$ is a constrained Nash equilibrium of $\Gamma^\epsilon$.

Let $\epsilon^{(n)}$ be a sequence so that $\epsilon^{(n)} > 0$ and $\epsilon^{(n)} \to 0$. Let $x_\star^{(n)} \in \mathcal{C}'$ be a constrained Nash equilibrium of $\Gamma^{\epsilon^{(n)}}$; this was just shown to exist. Let $x^\star$ be a limit point of the sequence $x_\star^{(n)}$, which exists as $x_\star^{(n)} \in \mathcal{C}'$ and $\mathcal{C}'$ is compact. For any $i \in [m]$, let $x_i' \in \mathcal{C}_i(x_{-i}^\star)$. Again, by playerwise concavity of constraints, $(x_i', x_{-i}^\star) \in \mathcal{C}'$. By continuity of $u_i$, and using the fact that $x_\star^{(n)}$ is a constrained Nash equilibrium of the perturbed game $\Gamma^{\epsilon^{(n)}}$, we have

$$u_i(x_i', x^\star) = \lim_{n \to \infty} u_i^{\epsilon^{(n)}}(x_i', x_\star^{(n)}) \leq \lim_{n \to \infty} u_i^{\epsilon^{(n)}}(x_\star^{(n)}) = u_i(x^\star),$$

i.e., we have shown that $x^\star$ is a constrained Nash equilibrium of the original unperturbed game. $\qquad\square$

# E. Background for Section 4

In this section, we introduce some background on constrained optimization relevant for the convergence analyses in Appendix F. First, we establish connections between KKT conditions and constrained Nash equilibria. Second, we prove a technical lemma which will be applied to establish a lower bound on the adaptive stepsizes.

## E.1. Lagrangian, KKT Conditions, and Connections to Constrained Nash Equilibria

We define Lagrangian functions in a playerwise manner, i.e., with respect to a fixed strategy for all except one player. Formally, for $i \in [m]$ and $x_{-i} \in \mathcal{X}_{-i}$, we define $\mathcal{L}_i^{x_{-i}} : \mathcal{X}_i \times \mathbb{R}^b \to \mathbb{R}$ as

$$\mathcal{L}_i^{x_{-i}}(x_i, \lambda) := u_i(x_i, x_{-i}) + \sum_{j \in [b]} \lambda_j \left( c_j(x_i, x_{-i}) - \alpha_j \right).$$

Next, we introduce approximate KKT conditions that are formulated based on the above Lagrangian in a playerwise manner.

**Definition E.1.** For $\epsilon > 0$, a strategy profile $x \in \mathcal{X}$ satisfies our $\epsilon$-KKT conditions, if there exist Lagrange multipliers $\lambda = (\lambda_1, \ldots, \lambda_b) \in \mathbb{R}^b$ such that

$$x \in \mathcal{C} \qquad \text{(primal feasibility)}$$

$$\min_{j \in [b]} \lambda_j \geq 0 \qquad \text{(dual feasibility)}$$

$$\max_{j \in [b]} |\lambda_j \left( c_j(x) - \alpha_j \right)| \leq \epsilon \qquad \text{(complementary slackness)}$$

$$\forall i \in [m], \quad \max_{x_i' \in \mathcal{X}_i} \langle x_i' - x_i, \nabla_{x_i} \mathcal{L}_i^{x_{-i}}(x_i, \lambda) \rangle \leq \epsilon \qquad \text{(Lagrangian stationarity)}$$

Next, we establish the following connection between these approximate KKT conditions and approximate constrained Nash equilibria.

**Lemma E.2.** *Let $\epsilon > 0$ and let $(x, \lambda) \in \mathcal{X} \times \mathbb{R}^b$ satisfy the above $\epsilon$-KKT conditions. Then $x$ is a constrained $(2\epsilon)$-approximate Nash equilibrium.*

*Proof.* Let $i \in [m]$ and $x_i' \in \mathcal{C}_i(x_{-i})$ be any feasible deviation of player $i$. Then, using concavity of $u_i$ and the definition of the Lagrangian, we bound

$$
\begin{aligned}
u_i(x_i', x_{-i}) - u_i(x_i, x_{-i}) &\leq \langle x_i' - x_i, \nabla_{x_i} u_i(x_i, x_{-i}) \rangle \\
&= \langle x_i' - x_i, \nabla_{x_i} \mathcal{L}_i^{x_{-i}}(x_i) \rangle - \sum_{j \in [b]} \lambda_j \langle x_i' - x_i, \nabla_{x_i} c_j(x_i, x_{-i}) \rangle .
\end{aligned}
\tag{3}
$$

For each $j \in [b]$, we can bound the respective term in the sum on the right-hand side using concavity of $c_j$ by

$$
\begin{aligned}
\lambda_j \langle x_i' - x_i, \nabla_{x_i} c_j(x_i, x_{-i}) \rangle &\geq \lambda_j c_j(x_i', x_{-i}) - \lambda_j c_j(x_i, x_{-i}) \\
&= \lambda_j \left( c_j(x_i', x_{-i}) - \alpha_j \right) - \lambda_j \left( c_j(x_i, x_{-i}) - \alpha_j \right) \\
&\geq -\lambda_j \left( c_j(x_i, x_{-i}) - \alpha_j \right)
\end{aligned}
\tag{4}
$$

where the last step is due to $x_i' \in \mathcal{C}_i(x_{-i})$, i.e., $c_j(x_i', x_{-i}) \geq \alpha_j$. Plugging (4) into (3), and applying Lagrangian stationarity and complementary slackness, we get

$$
u_i(x_i', x_{-i}) - u_i(x_i, x_{-i}) \leq \langle x_i' - x_i, \nabla_{x_i} \mathcal{L}_i^{x_{-i}}(x_i) \rangle + \sum_{j \in [b]} \lambda_j \left( c_j(x_i, x_{-i}) - \alpha_j \right) \leq 2\epsilon
$$

which implies the claimed statement by definition of approximate constrained Nash equilibrium. $\square$

### E.2. Technical Lemma

**Notation** For $x, x' \in \mathbb{R}^n$, we denote the line segment connecting $x$ and $x'$ by

$$
[x, x'] := \{ \lambda x + (1 - \lambda) x' \mid \lambda \in [0, 1] \} .
$$

We provide a lemma which is applied in the proof of Lemma F.8 in Appendix F. Informally, it says the following: If we perform a projected gradient ascent step along a (locally) smooth function $f$ and the observed gradients of $f$ are perturbed by some error $\delta$, then, as long as $\delta$ is sufficiently small, the value of $f$ will not decrease.

**Lemma E.3.** *Let $\Omega \subset \mathbb{R}^n$ and let $f : \Omega \to \mathbb{R}$ be differentiable. For $\kappa \geq 1$, let*

$$
\begin{aligned}
x' &:= x + \frac{1}{\kappa} \nabla f(x), \\
x^+ &:= \mathcal{P}_\Omega \left[ x' \right],
\end{aligned}
$$

*and define the mapping $F(x) := \kappa(x^+ - x)$. Suppose for all $\hat{x} \in [x, x^+]$, it holds that*

$$
\left\| \nabla f(x^+) - \nabla f(x) \right\| \leq \kappa \| x^+ - x \|.
$$

*For $\delta \in \mathbb{R}^n$, define the perturbed update as*

$$
\begin{aligned}
\widetilde{x}' &:= x + \frac{1}{\kappa} \left( \nabla f(x) + \delta \right), \\
\widetilde{x}^+ &:= \mathcal{P}_\Omega \left[ \widetilde{x}' \right].
\end{aligned}
$$

*If $\| F(x) \| \geq 3 \| \delta \|$, then $f(\widetilde{x}^+) \geq f(x)$.*

*Proof.* Let $\widetilde{F}(x) := \kappa \left( \widetilde{x}^+ - \widetilde{x} \right)$. We divide the proof into two steps.

- **Step 1:** First, we show that $\|\widetilde{F}(x)\| \geq 2\|\delta\|$. Namely, we can bound

$$
\|\widetilde{F}(x)\| \overset{a)}{\geq} \|F(x)\| - \|\widetilde{F}(x) - F(x)\|
$$

$$
\overset{b)}{\geq} 3\|\delta\| - \kappa\|\widetilde{x}^+ - x^+\|
$$

$$
\overset{c)}{\geq} 3\|\delta\| - \kappa\|\widetilde{x}' - x'\|
$$

$$
\overset{d)}{=} 3\|\delta\| - \|\delta\|
$$

$$
= 2\|\delta\|.
$$

where a) is by the triangle inequality, b) by assumption of the lemma and definition of $\widetilde{F}$ and $F$, c) by non-expansiveness of projection, and d) by definition of the perturbed update.

- **Step 2:** By the projection theorem, for all $z \in \Omega$, it holds that $(\widetilde{x}' - \widetilde{x}^+)^\top (z - \widetilde{x}^+) \leq 0$. Thus, with $z = x$ and $\widetilde{x}^+ - x = \frac{1}{\kappa}\widetilde{F}(x)$, we get

$$
(\widetilde{x}' - x + x - \widetilde{x}^+)^\top (z - \widetilde{x}^+) = \left(\frac{1}{\kappa}(\nabla f(x) + \delta) - \frac{1}{\kappa}\widetilde{F}(x)\right)^\top \left(-\frac{1}{\kappa}\widetilde{F}(x)\right) \leq 0.
$$

After rearranging, this gives

$$
\nabla f(x)^\top \widetilde{F}(x) \geq -\delta^\top \widetilde{F}(x) + \|\widetilde{F}(x)\|^2. \tag{5}
$$

Then, starting from the $\kappa$-smoothness lower bound, we get

$$
f(\widetilde{x}^+) \geq f(x) + \nabla f(x)^\top (\widetilde{x}^+ - x) - \frac{\kappa}{2}\|\widetilde{x}^+ - x\|^2
$$

$$
\overset{a)}{\geq} f(x) + \frac{1}{\kappa}\nabla f(x)^\top \widetilde{F}(x) - \frac{1}{2\kappa}\|\widetilde{F}(x)\|^2
$$

$$
\overset{b)}{\geq} f(x) - \frac{1}{\kappa}\delta^\top \widetilde{F}(x) + \frac{1}{2\kappa}\|\widetilde{F}(x)\|^2,
$$

$$
\overset{c)}{\geq} f(x) - \frac{1}{\kappa}\|\delta\| \cdot \|\widetilde{F}(x)\| + \frac{1}{2\kappa}\|\widetilde{F}(x)\|^2
$$

$$
\overset{d)}{\geq} f(x) - \frac{1}{2\kappa}\|\widetilde{F}(x)\|^2 + \frac{1}{2\kappa}\|\widetilde{F}(x)\|^2
$$

$$
\geq f(x)
$$

where a) is by definition of $\widetilde{F}$, b) uses inequality (5), c) is by the Cauchy-Schwarz inequality, and d) by the bound shown in **Step 1**.

$\square$

We restate the following sufficient increase inequality for projected gradient ascent that will be used in the proof of Theorem 4.7. The claimed inequality is standard; for reference, see Bubeck et al. (2015), Lemma 3.6.

**Lemma E.4.** *Let $\Omega \subset \mathbb{R}^n$ and let $f : \Omega \to \mathbb{R}$ be differentiable. For $\kappa \geq 1$, let*

$$
x^+ := \mathcal{P}_\Omega \left[ x + \frac{1}{\kappa}\nabla f(x) \right],
$$

*and suppose for all $\hat{x} \in [x, x^+]$,*

$$
\|\nabla f(\hat{x}) - \nabla f(x)\| \leq \kappa\|\hat{x} - x\|.
$$

*Then, we have*

$$
f(x^+) - f(x) \geq \frac{\kappa}{2}\|x - x^+\|^2.
$$

# F. Proofs for Section 4

In this section, we provide the full proofs for the results of Section 4.

## F.1. Mixed-Strategy Potential Games

In Example 4.2 we claimed that when considering mixed strategies over a finite action potential game, the potential structure is lifted to the continuous action game over compact convex set of mixed strategies. Here we state the corresponding proposition.

**Proposition F.1.** *Let $\mathcal{A} := \times_{i \in [m]} \mathcal{A}_i$ with each $\mathcal{A}_i$ finite, and suppose $\widetilde{u}_i : \mathcal{A} \to [0, 1]$ are player utilities and $\widetilde{c}_j : \mathcal{A} \to [0, 1]$ constraints with thresholds $\alpha_j$ inducing feasible region $\widetilde{\mathcal{C}}$, such that there exists a potential function $\phi : \widetilde{\mathcal{C}} \to \mathbb{R}$ satisfying*

$$\phi(a) - \phi(a_i', a_{-i}) = \widetilde{u}_i(a) - \widetilde{u}_i(a_i', a_{-i}) \quad \text{for all } i \in [m], \, a \in \widetilde{\mathcal{C}}, \, a_i' \in \widetilde{\mathcal{C}}_i(a_{-i}).$$

*Then, the continuous action game with strategy space $\mathcal{X} := \times_{i \in [m]} \Delta(\mathcal{A}_i)$, utilities*

$$u_i(x) := \mathbb{E}_{a \sim x}[\widetilde{u}_i(a)]$$

*and constraints*

$$c_j(x) := \mathbb{E}_{a \sim x}[\widetilde{c}_j(a)]$$

*with thresholds $\alpha_j$ is a constrained potential game with potential function $\Phi(x) := \mathbb{E}_{a \sim x}[\phi(a)]$.*

For the case of unconstrained potential games, Proposition F.1 is a direct consequence of Monderer & Shapley (1996), Lemma 2.10. The same reasoning extends straightforwardly to the constrained case; we therefore omit the proof here.

## F.2. Potential Game Structure

Next, we provide two lemmas on the potential structure and properties of the potential function that will enable our convergence analysis. First, we show that the log barrier regularized game is still a potential game.

**Lemma F.2.** *Suppose $\Phi : \mathcal{X} \to \mathbb{R}$ is a potential function for the game with utilities $\{u_i\}_{i \in [m]}$. Then $\Phi^{\eta}(x) := \Phi(x) + \eta \sum_{j \in [b]} \log(c_j(x) - \alpha_j)$ is a potential for the log barrier regularized game with utilities $\{B_i^{\eta}\}_{i \in [m]}$.*

*Proof.* If we have a game with utilities $\{u_i'\}_{i \in [m]}$ and potential function $\Phi'$, and another game with utilities $\{u_i''\}_{i \in [m]}$ and potential function $\Phi''$, it is immediate that then $\Phi' + \Phi''$ is a potential function for the game with utilities $\{u_i' + u_i''\}_{i \in [m]}$. The statement hence follows by observing that the sum of log barrier terms forms a cooperative game and hence $\eta \sum_{j \in [b]} \log(c_j(x) - \alpha_j)$ is a potential function for it. $\square$

Next, we show that the potential function of the unregularized game is smooth.

**Lemma F.3.** *Let $\Phi : \mathcal{X} \to \mathbb{R}$ be a potential function for our game with utilities $\{u_i\}_{i \in [m]}$. Then $\Phi$ is $(M\sqrt{m})$-smooth in the joint strategy $x \in \mathcal{X}$, where $M$ is the smoothness parameter of the utilities, see Assumption 4.4.*

*Proof.* Note that differentiability of $u_i$ for all $i \in [m]$ (Assumption 4.4) implies differentiability of $\Phi$, and it is well known that for all $x \in \mathcal{X}$, $\nabla_{x_i} \Phi(x) = \nabla_{x_i} u_i(x)$. Using the smoothness of $u_i$ from Assumption 4.4, we get

$$\|\nabla\Phi(x) - \nabla\Phi(x')\|^2 = \sum_{i \in [m]} \|\nabla_{x_i}\Phi(x) - \nabla_{x_i}\Phi(x')\|^2$$

$$= \sum_{i \in [m]} \|\nabla_{x_i}u_i(x) - \nabla_{x_i}u_i(x')\|^2$$

$$\leq M^2 \sum_{i \in [m]} \|x - x'\|^2$$

$$\leq M^2 m \|x - x'\|^2$$

from which we conclude that $\|\nabla\Phi(x) - \nabla\Phi(x')\| \leq M\sqrt{m}\|x - x'\|$. $\square$

### F.3. Smoothness Along the Trajectory

In the following subsections, we present a convergence analysis of our independent log barrier method applied to constrained games with potential structure. We begin by examining the smoothness of properties of $\Phi^\eta$ along the joint gradient ascent trajectory. For this, we introduce the following notion of $\beta$-feasibility.

**Definition F.4.** A strategy $x \in \mathcal{C}$ is $\beta$-feasible for $\beta \geq 0$, if $c_j(x) - \alpha_j \geq \beta$ for all $j \in [b]$.

First, we show that $\beta$-feasibility can be preserved along the line segment spanned by consecutive iterates generated by update (1), here up to a factor of $1/2$, when choosing sufficiently small stepsizes.

**Lemma F.5.** Let $x^{(t)} \in \mathcal{C}$ be $\beta$-feasible. If $\gamma^{(t)} \leq \frac{\beta^2}{2mL^2(\beta+\eta b)}$, then for all $\hat{x} \in [x^{(t)}, x^{(t+1)}]$, $\hat{x}$ is $\beta/2$-feasible.

*Proof.* First, we use $\beta$-feasibility of $x^{(t)}$ to bound the norm of log barrier regularized partial gradients. Namely, for any $i \in [m]$, we have

$$\|\nabla_{x_i} B_i^\eta(x^{(t)})\| \leq \|\nabla_{x_i} u_i(x^{(t)})\| + \eta \sum_{j \in [b]} \frac{\|\nabla_{x_i} c_j(x^{(t)})\|}{\beta} \leq \left(1 + \frac{\eta b}{\beta}\right) L = \frac{(\beta + \eta b)L}{\beta}$$

where we use $L$-Lipschitz continuity of $u_i$ and $c_j$, see Assumption 4.4, for bounding the gradient norms. Then, we have for each $j \in [b]$,

$$\begin{aligned}
c_j(\hat{x}) - \alpha_j &\geq c_j(x^{(t)}) - \alpha_j - L\|\hat{x} - x^{(t)}\| \\
&\geq c_j(x^{(t)}) - \alpha_j - L\|x^{(t+1)} - x^{(t)}\| \\
&\geq c_j(x^{(t)}) - \alpha_j - L \sum_{i \in [m]} \|x_i^{(t+1)} - x_i^{(t)}\| \\
&\overset{a)}{\geq} c_j(x^{(t)}) - \alpha_j - L \sum_{i \in [m]} \gamma^{(t)} \|\nabla_{x_i} B_i^\eta(x^{(t)})\| \\
&\overset{b)}{\geq} c_j(x^{(t)}) - \alpha_j - \frac{\beta}{2} \\
&\overset{c)}{\geq} \frac{c_j(x^{(t)}) - \alpha_j}{2}
\end{aligned}$$

where a) is by non-expansiveness of the projection operator, b) by our choice of $\gamma^{(t)}$, and c) by $\beta$-feasibility of $x^{(t)}$. $\qquad\square$

For $\beta$-feasible strategies, we give the following Hessian norm bound.

**Lemma F.6.** Let $x \in \mathcal{C}$ be $\beta$-feasible. Then $\|\nabla^2 \Phi^\eta(x)\| \leq M_\Phi^{\eta,\beta}$ where

$$M_\Phi^{\eta,\beta} := M\sqrt{m} + \eta M b \beta^{-1} + \eta L^2 b \beta^{-2}.$$

*Proof.* For any $x \in \mathcal{X}$, we have

$$\nabla^2 \Phi^\eta(x) = \nabla^2 \Phi(x) - \eta \sum_{j \in [b]} \frac{\nabla^2 c_j(x)}{c_j(x) - \alpha_j} + \eta \sum_{j \in [b]} \frac{\nabla c_j(x) \nabla c_j(x)^\top}{(c_j(x) - \alpha_j)^2}.$$

Using $M$-smoothness and $L$-Lipschitz continuity of $c_j$ for all $j \in [b]$, as well as $(M\sqrt{m})$-smoothness of $\Phi$, as shown in Lemma F.3, we then bound the Hessian norm as follows:

$$\begin{aligned}
\|\nabla^2 \Phi^\eta(x)\| &\leq \|\nabla^2 \Phi(x)\| + \eta \sum_{j \in [b]} \frac{\|\nabla^2 c_j(x)\|}{c_j(x) - \alpha_j} + \eta \sum_{j \in [b]} \frac{\|\nabla c_j(x) \nabla c_j(x)^\top\|}{(c_j(x) - \alpha_j)^2} \\
&\leq M\sqrt{m} + \eta \sum_{j \in [b]} \frac{M}{c_j(x) - \alpha_j} + \eta \sum_{j \in [b]} \frac{L^2}{(c_j(x) - \alpha_j)^2} \\
&\leq M\sqrt{m} + \eta M b \beta^{-1} + \eta L^2 b \beta^{-2} \\
&= M_\Phi^{\eta,\beta}.
\end{aligned}$$

$\square$

The Hessian norm bound from above is used to establish smoothness along the trajectory as follows.

**Lemma F.7.** *Suppose $x^{(t)} \in \mathcal{C}$ is $\beta$-feasible and each $\gamma^{(t)}$ is bounded as in the statement of Lemma F.5. Then $\Phi^{\eta}$ is $M_{\Phi}^{\eta,\beta/2}$-smooth over $[x^{(t)}, x^{(t+1)}]$ where $M_{\Phi}^{\eta,\beta}$ is defined as in Lemma F.6, i.e. for all $x, x' \in [x^{(t)}, x^{(t+1)}]$,*

$$\|\nabla\Phi^{\eta}(x) - \nabla\Phi^{\eta}(x')\| \leq M_{\Phi}^{\eta,\beta/2}\|x - x'\|.$$

*Proof.* By the fundamental theorem of calculus,

$$\nabla\Phi(x) - \nabla\Phi(x') = \int_0^1 \nabla^2\Phi(x' + \tau(x - x'))(x - x')\,d\tau.$$

Taking the norm, and using the fact that $x' + t(x - x')$ is $\beta/2$-feasible for all $t \in [0, 1]$ as shown in Lemma F.5, together with the Hessian norm bound from Lemma F.6, we get

$$
\begin{aligned}
\|\nabla\Phi^{\eta}(x) - \nabla\Phi^{\eta}(x')\| &\leq \int_0^1 \|\nabla^2\Phi^{\eta}(x' + \tau(x - x'))\| \cdot \|x - x'\|\,d\tau \\
&\leq \int_0^1 M_{\Phi}^{\eta,\beta/2}\|x - x'\|\,d\tau \\
&\leq M_{\Phi}^{\eta,\beta/2}\|x - x'\|.
\end{aligned}
$$

$\square$

### F.4. Lower Bounding the Stepsize

Motivated by the bound in Lemma F.5 and smoothness along the trajectory as shown in Lemma F.7, we choose the stepsize at iteration $t$ as

$$\gamma^{(t)} := \min\left\{ \frac{\beta^{(t)^2}}{2mL^2(\beta^{(t)} + \eta b)}, \frac{1}{M_{\Phi}^{\eta,\beta^{(t)}/2}} \right\} \tag{6}$$

where $\beta^{(t)} := \min_{j \in [b]}\{c_j(x^{(t)}) - \alpha_j\}$, i.e. such that $x^{(t)}$ is $\beta^{(t)}$-feasible[4].

In this subsection, we want to argue that the above defined stepsizes can be uniformly lower bounded by a constant multiple of $\eta$. This is necessary to guarantee sufficient progress in our convergence analysis. We start by showing that feasibility margins do not decrease further once a point sufficiently close to the boundary is reached. Based on Lemma E.3, we prove the following statement. Note that a similar statement was proven in Usmanova et al. (2024), Fact 1, with the difference that ours is for projected gradient steps, which requires a different approach.

In the following, for $j \in [b]$, we write $\beta_j^{(t)} := c_j(x^{(t)}) - \alpha_j$.

**Lemma F.8.** *Let Assumption 4.5 hold with $\rho \geq \eta$. If at any iteration $0 \leq t \leq T - 1$, we have $\beta^{(t)} \leq \bar{c}\eta$ with $\bar{c} = \frac{\ell}{6L(b+1)}$, then for iteration $t + 1$, we get $\prod_{j \in \mathcal{J}} \beta_j^{(t+1)} \geq \prod_{j \in \mathcal{J}} \beta_j^{(t)}$ for any $\mathcal{J}$ such that $\mathcal{J}_t \subseteq \mathcal{J}$ with $\mathcal{J}_t := \{j \in [b] \mid \beta_j^{(t)} \leq \eta\}$.*

*Proof.* We aim to apply Lemma E.3 with $x = x^{(t)}, x^+ = x^{(t+1)}$,

$$\delta = \nabla\Phi(x^{(t)}) + \eta \sum_{j \in [m] \setminus \mathcal{J}_t} \nabla\log\beta_j^{(t)}, \text{ and}$$

$$f(x) = \eta \sum_{j \in \mathcal{J}_t} \log\beta_j^{(t)}.$$

---

[4]Note that stepsizes (6) differ from what is specified as (2) in the main part of the paper. This is a typo and will be fixed in a later version.

Note that by our choice of $\gamma^{(t)}$, by Lemma F.7, we have $\|\nabla f(x^+) - \nabla f(x)\| \leq \frac{1}{\gamma^{(t)}}\|x^+ - x\|$. In order to apply Lemma E.3, we further need to show that for $F(x) := \frac{1}{\gamma^{(t)}}(x^+ - x)$, it holds that $\|F(x)\| \geq 3\|\delta\|$.

We begin by giving a lower bound on $\|F(x^{(t)})\|$. By Proposition B.1 of Agarwal et al. (2021) and Assumption 4.5, we have

$$
\begin{aligned}
2\left\|F(x^{(t)})\right\| &\geq \max_{x^{(t)}+d \in \mathcal{X}, \|d\| \leq 1} d^\top \nabla f(x^{(t)}) \\
&\geq \eta \max_{x^{(t)}+d \in \mathcal{X}, \|d\| \leq 1} \sum_{j \in \mathcal{J}_t} \frac{d^\top \nabla c_j(x^{(t)})}{\beta_j^{(t)}} \\
&\geq \eta \sum_{j \in \mathcal{J}_t} \frac{\ell}{\beta_j^{(t)}} \\
&\geq \frac{\ell}{\bar{c}}.
\end{aligned}
$$

Then, we show an upper bound on $\|\delta\|$,

$$
\begin{aligned}
\|\delta\| &= \left\|\nabla \Phi(x^{(t)}) + \eta \sum_{j \in [m] \setminus \mathcal{J}_t} \nabla \log \beta_j^{(t)}\right\| \\
&\leq \left\|\nabla \Phi(x^{(t)})\right\| + \eta \sum_{j \in [m] \setminus \mathcal{J}_t} \left\|\frac{\nabla c_j(x^{(t)})}{\beta_j^{(t)}}\right\| \\
&\overset{(\star)}{\leq} L + \eta \cdot b \cdot \frac{L}{\eta} \\
&\leq L(b+1),
\end{aligned}
$$

where $(\star)$ is due to $\beta_j^{(t)} > \eta$ for $j \notin \mathcal{J}_t$. Combining the two bounds, and by definition of $\bar{c}$, we obtain from Lemma E.3 that

$$
\eta \sum_{j \in \mathcal{J}_t} \log \beta_j^{(t+1)} \geq \eta \sum_{j \in \mathcal{J}_t} \log \beta_j^{(t)}.
$$

Moreover, using the same reasoning, we can prove this inequality to hold for the summation over $\mathcal{J}$ instead of $\mathcal{J}_t$. Then, dividing by $\eta$ and exponentiating both sides yields the claimed result. $\qquad \square$

As a consequence of Lemma F.8, we obtain the following lower bound on the distance of iterates to the boundary of $\mathcal{C}$. Having established Lemma F.8 above, the proof of Lemma F.9 is equivalent to the proof of an analogous result stated as Lemma 6 in Usmanova et al. (2024); we omit the proof here.

**Lemma F.9.** *Let Assumptions 4.3 and 4.5 hold with $\rho \geq \eta$. Then, for all $t \geq 1$, it holds that $\beta^{(t)} \geq c\eta$ where $c := \frac{1}{2}\left(\frac{\ell}{24L(b+1)}\right)^b$.*

The above lower bound also implies a lower bound on the stepsizes.

**Lemma F.10.** *If $\beta^{(t)} \geq c\eta$ for some $c > 0$, then we have $\gamma^{(t)} \geq C\eta$ with*

$$
C := \min\left\{\frac{c}{2mL^2(1+b)}, \frac{1}{M\sqrt{m} + 2Mbc^{-1} + 4L^2bc^{-2}}\right\}
$$

*Proof.* Recall that we have set

$$
\gamma^{(t)} := \min\left\{\frac{\beta^{(t)2}}{2mL^2(\beta^{(t)} + \eta b)}, \frac{1}{M_\Phi^{\eta,\beta^{(t)}/2}}\right\}.
$$

As we are interested in small $\eta > 0$, we may assume w.l.o.g. that $\eta \leq 1$. We may further assume that $\beta^{(t)} \leq 1$; otherwise, we trivially have a lower bound on $\beta^{(t)}$. Then the first argument of the minimum above is lower bounded by

$$\frac{\beta^{(t)^2}}{2mL^2(\beta^{(t)} + \eta b)} \geq \frac{\beta^{(t)}}{2mL^2(1 + \eta b)} \geq \frac{\eta c}{2mL^2(1 + b)}$$

For the second term, by definition of $M_\Phi^{\eta,\beta}$, we get

$$\frac{1}{M_\Phi^{\eta,\beta^{(t)}/2}} \geq \frac{1}{M\sqrt{m} + \frac{2Mb}{c} + \frac{4L^2b}{c^2\eta}}$$

$$\geq \frac{\eta}{\eta M\sqrt{m} + \frac{2\eta Mb}{c} + \frac{4L^2b}{c^2}}$$

$$\geq \frac{\eta}{M\sqrt{m} + \frac{2Mb}{c} + \frac{4L^2b}{c^2}}$$

The claimed bound thus follows by combining both. $\square$

Finally, we are ready to prove the main convergence result for the constrained potential game setting.

**Theorem 4.7.** *Let $\epsilon > 0$ and set $\eta = \epsilon$. Let Assumptions 2.2, 4.3, 4.4, and 4.5 hold with $\rho \geq \eta$. For stepsizes $\{\gamma^{(t)}\}_{t=0}^{T-1}$ chosen as in (2), suppose we perform $T = \mathcal{O}(\epsilon^{-3})$ rounds of simultaneous updates as in (1). Then, there exists $0 \leq t \leq T$ such that $x^{(t)}$ is a constrained $\epsilon$-approximate Nash equilibrium.*

*Proof.* First, we argue that our independent updates are equivalent to centralized gradient ascent on the regularized potential function $\Phi^\eta$. This follows from the fact that, by Lemma F.2, the regularized utilities form a potential game, meaning we have $\nabla_{x_i}\Phi^\eta(x) = \nabla_{x_i}B_i^\eta(x)$ for any $x \in \mathcal{X}$ and $i \in [m]$. Together with separability of the projection operator onto the Cartesian product of the sets $\mathcal{X}_i$, see Leonardos et al. (2022), Lemma D.1, we obtain

$$\left(\mathcal{P}_{\mathcal{X}_i}\left[x_i^{(t)} + \gamma^{(t)}\nabla_{x_i}B_i^\eta(x^{(t)})\right]\right)_{i\in[m]} = \mathcal{P}_{\mathcal{X}}\left[x^{(t)} + \gamma^{(t)}\nabla\Phi^\eta(x^{(t)})\right].$$

Hence, in the following we carry out the analysis by viewing the algorithm as a centralized gradient ascent on $\Phi^\eta$.

By smoothness along the trajectory and our choice of stepsizes, by Lemma E.4, for all $0 \leq t \leq T - 1$,

$$\Phi^\eta(x^{(t+1)}) - \Phi^\eta(x^{(t)}) \geq \frac{\gamma^{(t)}}{2}\left\|x^{(t+1)} - x^{(t)}\right\|^2.$$

Together with the fact that there exists a constant $C > 0$ such that $\gamma^{(t)} \geq C\eta$, see Lemma F.10, and by telescoping the sum, we obtain

$$\frac{1}{T}\sum_{t=0}^{T-1}\left\|x^{(t+1)} - x^{(t)}\right\|^2 \leq \frac{2}{T}\sum_{t=0}^{T-1}\frac{\Phi^\eta(x^{(t+1)}) - \Phi^\eta(x^{(t)})}{\gamma^{(t)}}$$

$$\leq \frac{2\Phi^\eta(x^{(T)})}{C\eta T}.$$

We note that using Lemma F.9, $\Phi^\eta$ can be bounded by a constant. Thus for $\eta = \epsilon$ and $T = \mathcal{O}(\epsilon^{-3})$, there exists $0 \leq t^\star \leq T - 1$ such that $\|x^{(t^\star+1)} - x^{(t^\star)}\| \leq \frac{\epsilon}{4}$. Next, we argue that then $x^{(t^\star+1)}$ satisfies the $\frac{\epsilon}{2}$-KKT conditions as introduced in Appendix E. Consider the pair of primal and dual variables given by

$$(x^\star, \lambda^\star) := \left(x^{(t^\star+1)}, \frac{\eta}{2}\left[(c_j(x^{(t^\star+1)}) - \alpha_j)^{-1}\right]_{j\in[b]}\right).$$

We check each of the conditions:

1. **Primal feasibility:** Lemma F.5 implies that all iterates remain within $\mathcal{C}$. In particular, we have $x^\star \in \mathcal{C}$.

2. **Dual feasibility:** This follows directly from primal feasibility which implies that $c_j(x^\star) - \alpha_j \geq 0$ for all $j \in [b]$, and thus $\lambda_j^\star \geq 0$.

3. **Complementary slackness:** This is a consequence of the definition of $\lambda^\star$, as

$$\max_{j \in [b]} \left| \lambda_j^\star (c_j(x^\star) - \alpha_j) \right| = \frac{\eta}{2} = \frac{\epsilon}{2}.$$

4. **Lagrangian stationarity:** By a standard property of projected gradient ascent, see e.g. Agarwal et al. (2021), Proposition B.1, $\|x^{(t^\star+1)} - x^{(t^\star)}\| \leq \frac{\epsilon}{4}$ implies that

$$\max_{x \in \mathcal{X}} \langle x - x^\star, \nabla \Phi^\eta(x^\star) \rangle \leq \frac{\epsilon}{2}.$$

Then we can bound, for any $i \in [m]$,

$$\begin{aligned}
\max_{x_i \in \mathcal{X}_i} \left\langle x_i^\star - x_i, \nabla_{x_i} \mathcal{L}_i^{x_{-i}^\star}(x_i^\star, \lambda^\star) \right\rangle &= \max_{x_i \in \mathcal{X}_i} \left\langle x^\star - (x_i, x_{-i}^\star), \nabla \Phi^\eta(x^\star) \right\rangle \\
&\leq \max_{x \in \mathcal{X}} \left\langle x^\star - x, \nabla \Phi^\eta(x^\star) \right\rangle \\
&\leq \frac{\epsilon}{2},
\end{aligned}$$

where the first step is by definition of $\lambda^\star$.

Finally, Lemma E.2 implies that the $\frac{\epsilon}{2}$-KKT strategy $x^\star$ constitutes an $\epsilon$-approximate constrained Nash equilibrium. □

