# OpenReview forum: "Nash Equilibria in Games with Playerwise Concave Coupling Constraints: Existence and Computation"
_ICML.cc/2026/Conference — ICML 2026 spotlight_

### Official Review · Reviewer_7ciu · 2026-02-26

**Soundness:** 3
**Presentation:** 3
**Significance:** 3
**Originality:** 3
**Overall Recommendation:** 5
**Confidence:** 3

**Summary:**

This paper studies a class of continuous games where players' utilities are playerwise concave and the players' strategies are subject to shared coupling constraints. First, the authors show that under appropriate assumptions on the feasible strategies, a constrained Nash equilibrium always exists. The proof of this result establishes the contractibility over connected components of the feasible region, using topological arguments. Computationally, the paper studies log-barrier regularized gradient ascent, a multiagent/independent version of an interior point optimization algorithm. The algorithm is shown to exhibit $O(\epsilon^{-3})$ convergence rate to an $\epsilon$-approximate Nash equilibrium. Finally, empirical evidence is shown that the proposed algorithm indeed converges in terms of exploitability to approximate Nash equilibria.

**Compliance With Llm Reviewing Policy:**

Affirmed.

**Final Justification:**

I think this is a good paper with a significant contribution. My concerns were fully resolved and I am will maintain my score and support of the paper.

**Key Questions For Authors:**

- For the convergence result, you have used a relation between Nash equilibria and KKT points in potential games. However, to my understanding, beyond potential games KKT and Nash might no longer coincide. Is this the main barrier towards obtaining convergence results for zero-sum/general-sum games? Experimentally, for the log-barrier regularized gradient ascent algorithm, do you observe approximate convergence to equilibria in these games?

**Limitations:**

Yes

**Strengths And Weaknesses:**

### Strengths
- The motivation to study games with shared coupling constraints is well argued for, and the structure of the paper nicely builds up from showing equilibrium existence to efficient computation in potential games.
- the existence result is interesting, and the techniques used for contractibility over nonconvex feasible regions will likely be of interest since they extend methods to show equilibrium existence beyond variational inequalities in the jointly convex case.
- The algorithm studied exhibits behavior which is reasonable given the nonconvexity of the game class (i.e. slower convergence than in constrained potential games without coupling constraints). The linear convergence rate (even in the case of potential games) is a significant result, especially considering the class of games has a possibly nonconvex feasible region.

### Weaknesses
- While the examples/experiments given in the main text are illustrative, an experiment on a more realistic game would make the contribution stronger. In particular, are there reasonable classes of shared constraints that arise from the settings described in the intro (e.g. collision avoidance, bandwidth allocation) that can be evaluated experimentally?
- Assumption 4.5 seems potentially restrictive, since it requires a uniform ascent direction everywhere near the boundary of the feasible space. Indeed, requiring an ascent direction for all constraints on every boundary is a restriction that already does not capture some simple games (cf. Example 3.2). Does Assumption 4.5 further shift the needle, i.e. concretely what are some settings of interest that this assumption still allows the game class to capture?

---

> ### Author Rebuttal · Authors · 2026-03-31
>
> We thank the reviewer for bringing up several interesting points and would like to address them as follows:
> 1. The particular connection we use between KKT conditions and constrained Nash equilibria (proven in Lemma E.2) is not specific to potential games. In our result, the potential structure is used mainly for showing convergence of our algorithm to an approximate KKT point. We also refer to related literature on variational inequalities and generalized Nash equilibrium problems where similar connections between KKT conditions and constrained games are made without restricting to any particular utility structure, see [1].
>
> 2. We believe the main challenge in generalizing our result to other game classes, such as zero-sum games, is as follows: If we take a potential game and add the log barrier terms to the utilities, the resulting regularized game is still potential (see our Lemma F.2). However, if we take a zero-sum game and add log barrier terms, then due to the constraints being shared, the resulting regularized game is no longer zero-sum. It is then no longer clear whether this regularized general-sum game has any other structural properties that makes Nash equilibrium finding tractable. This lack of structure appears to be the key obstacle to extending our approach beyond potential games.
>
> 3. Regarding your question on empirical convergence in zero-sum games, we have indeed conducted some numerical experiments in this setting. For illustration, [here](https://osf.io/hbuac/files/bt65y?view_only=cd6b4d6a99f54300bd997080414cf424) we plot the trajectory of a log barrier regularized optimistic gradient descent ascent (optimism to prevent cycling) in such a convex-concave saddle point problem with a shared coupling constraint. In these experiments we do observe convergence to constrained Nash equilibria. However, due to the general-sum structure induced by the regularization (as described in 2.), we would expect that counterexamples may exist in principle. However, so far, we have not been able to construct such instances. It therefore remains an interesting open question whether such non-convergent examples exist, or whether some other structure can be leveraged to establish convergence also in this constrained zero-sum case.
>
> 4. We particularly appreciate your comments on Assumption 4.5. After revisiting our proof and related literature, we have come to the conclusion that in our setting, due to compactness of strategy spaces and Lipschitz continuity of constraint functions, the non-extended version of our MFCQ condition (Assumption 3.1) directly implies existence of some $\ell > 0$ and $\rho > 0$ for which the extended MFCQ condition (Assumption 4.5) holds. Intuitively, this is because Lipschitz continuity prevents gradient directions from changing too quickly when moving from the boundary to the interior. A rigorous argument for this sufficient condition is given in Appendix B.1 of [2]. So, while the initial result under Assumption 4.5 was correct, for the final version of the paper, we will replace it with the strengthened version that relies only on the more common Assumption 3.1. Thank you for helping to point us to this valuable insight which we believe improves the clarity and applicability of our results.
>
> 5. Regarding more experiments on practical applications: Several of the mentioned scenarios such as multi-robot interactions [3] or power control problems [4] have been modelled as potential games with coupling constraints and would hence provide an interesting test bed for our methods. We aim to run such experiments in our future work.
>
> ---
>
> [1] Facchinei, Francisco, Andreas Fischer, and Veronica Piccialli. "On generalized Nash games and variational inequalities." Operations Research Letters 35, no. 2 (2007): 159-164.
>
> [2] Ni, Tingting, and Maryam Kamgarpour. "A Safe Exploration Approach to Constrained Markov Decision Processes." In International Conference on Artificial Intelligence and Statistics, pp. 3592-3600. PMLR, 2025.
>
> [3] Bhatt, Maulik, Yixuan Jia, and Negar Mehr. "Efficient constrained multi-agent trajectory optimization using dynamic potential games." In 2023 IEEE/RSJ International Conference on Intelligent Robots and Systems (IROS), pp. 7303-7310. IEEE, 2023.
>
> [4] Scutari, Gesualdo, Sergio Barbarossa, and Daniel P. Palomar. "Potential games: A framework for vector power control problems with coupled constraints." In 2006 IEEE International Conference on Acoustics Speech and Signal Processing Proceedings, vol. 4, pp. IV-IV. IEEE, 2006.

---

> > ### Author Rebuttal · Reviewer_7ciu · 2026-04-01
> >
> > The rebuttal address the concerns that I had, particularly on the restrictiveness of Assumption 4.5. The authors strengthening the argument and thus needing only Assumption 3.1 is a solid improvement to the significance of the results, and so I will maintain my score and support of the paper.

---

### Official Review · Reviewer_8eAf · 2026-03-10

**Soundness:** 3
**Presentation:** 3
**Significance:** 3
**Originality:** 3
**Overall Recommendation:** 5
**Confidence:** 3

**Summary:**

This paper studies games with coupled constraints. The authors first extend the existence results for Nash equilibria to settings with player-wise convex constraints. To establish this extension, the paper develops a contraction argument within each connected component of the induced feasibility set. The authors further study the computation of Nash equilibria in constrained potential games and propose an algorithm based with a log-barrier regularizer that achieves $(1/\epsilon^3)$ convergence. Finally, experimental results are provided to support the theoretical findings.

**Compliance With Llm Reviewing Policy:**

Affirmed.

**Final Justification:**

The authors addressed all of my concerns during the rebuttal period and I maintain my positive evaluation.

**Key Questions For Authors:**

- Some previous works studying the existence of Nash equilibria under coupled constraints often rely on a Slater condition [1]. Is there any connection between the player-wise convex constraint structure considered in this paper and the Slater condition?
- Compared to [2], the proposed algorithm incurs an additional $O(1/\epsilon)$ factor in the running time. The paper suggests that this is due to the use of a varying step size in the algorithm. Is there any evidence indicating whether this additional factor is unavoidable?

-----
[1] Altman, E. and Shwartz, A. Constrained Markov Games: Nash Equilibria. In Advances in Dynamic Games and Applications, Annals of Independent Learning in Constrained Markov Potential Games the International Society of Dynamic Games, 2000.

[2] Anagnostides, I., Panageas, I., Farina, G., and Sandholm, T. On last-iterate convergence beyond zero-sum games. In International Conference on Machine Learning, pp. 536–581. PMLR, 2022.

**Limitations:**

Yes

**Strengths And Weaknesses:**

Strengths:
- The coupled-constraint setting is fundamental. It is novel for the authors to extend the existence results for Nash equilibria to player-wise convex constraints, which can induce a non-convex joint feasibility domain.
- The paper proposes independent algorithm with simultaneous updates and provides theoretical guarantees on the convergence rates.
- Most of the assumptions made in the paper are well motivated.
- It is novel that the authors show that simultaneous gradient updates do not violate the constraints.
- Experimental results are included to support the theoretical findings.
- The paper is well written and includes detailed and rigorous proofs.

Weaknesses:
- The positive results are limited to constrained potential games. Other fundamental classes of games, such as zero-sum games, are not examined.
- Assumption 4.5 is not well motivated. Although it appears necessary for the analysis, the current statement does not make clear how strong or restrictive this assumption is.

---

> ### Author Rebuttal · Authors · 2026-03-31
>
> We appreciate the reviewer's positive feedback and their interesting questions, which we would like to address as follows:
> 1. Thanks for your questions concerning the work of Altman \& Shwartz [1]. We are indeed aware of this result and would like to clarify the connection with ours: Their strong Slater condition assumes (similarly to the result of Debreu that we discuss) that for *any* joint strategy of the other players, player $i$ can find a response that renders the joint strategy feasible. As we argue, this can be restrictive; for instance, two players may violate a collision-avoidance constraint that a third player cannot influence. Moreover, since the existence result of Altman \& Shwartz is established in a Markov game setting, our result does not directly apply, as the corresponding playerwise optimization problem (i.e., computing an optimal policy in an MDP) is generally nonconvex. Likewise, their result is not directly applicable to our game setting either, as we allow for general continuous utilities over compact domains that do not need to take the form of expected cumulative costs. This discussion seems insightful to the reader and we will include it as a remark in the final version of the paper.
>
> 2. Yes, compared to the corresponding result in games without coupling constraints of Anagnostides et al. [2], we pay an additional $\mathcal{O}(\epsilon^{-1})$ factor in the iteration complexity. We note that this is consistent with results on log barrier methods in nonconvex smooth constrained optimization: here the optimal rate for gradient descent ascent in the unconstrained case would be $\mathcal{O}(\epsilon^{-2})$ while the respective log barrier approach for the constrained case also takes $\mathcal{O}(\epsilon^{-3})$ iterations, see [3]. We are not aware of any lower bounds for such constrained optimization problems, neither in the single-player nor in the game setting.
>
> 3. Regarding the statement that game classes other than potential games are not examined, we would like to clarify that our *existence* result does not make any assumptions on the utility structure of the game, hence applies to any playerwise concave general-sum utilities. For the *learning* result, we agree that extending it to other game classes is an interesting direction for future work.
>
> 4. After receiving several remarks from reviewers about Assumption 4.5, we have revisited our result and related literature: It turns out that in our particular setting with compact strategy spaces and Lipschitz continuous constraint functions, this extended MFCQ condition is actually directly implied by Assumption 3.1, the non-extended version of our MFCQ condition (see also response to reviewer `7ciu`). The proof of this implication is contained in Appendix B.1 of [4]. Therefore, while the initial result under Assumption 4.5 was correct, for the final version of the paper, we will replace it with the strengthened version that relies only on the more common Assumption 3.1. We believe this improves the clarity and applicability of our result. In terms of motivating Assumption 3.1, we point out that assumptions of this type are commonly made in constrained game formulations, see for example [5].
>
> ---
>
> [3] Usmanova, Ilnura, Yarden As, Maryam Kamgarpour, and Andreas Krause. "Log barriers for safe black-box optimization with application to safe reinforcement learning." Journal of Machine Learning Research 25, no. 171 (2024): 1-54.
>
> [4] Ni, Tingting, and Maryam Kamgarpour. "A Safe Exploration Approach to Constrained Markov Decision Processes." In International Conference on Artificial Intelligence and Statistics, pp. 3592-3600. PMLR, 2025.
>
> [5] Dreves, Axel, Francisco Facchinei, Christian Kanzow, and Simone Sagratella. "On the solution of the KKT conditions of generalized Nash equilibrium problems." SIAM Journal on Optimization 21, no. 3 (2011): 1082-1108.

---

> > ### Author Rebuttal · Reviewer_8eAf · 2026-04-01
> >
> > I thank the authors for their rebuttal. I have one follow-up question.
> >
> > - In response to Reviewer 7ciu, the authors state that Assumption 4.5 directly follows from Assumption 3.1 in their setting. However, for Theorem 4.7 to hold, it is required that $\rho > \eta = \epsilon$. Is this condition also implied by Assumption 3.1 or is a separate assumption on $\rho$ still required?

---

> > > ### Author Response · Authors · 2026-04-02
> > >
> > > Thanks, yes, it is correct that our convergence guarantee requires $\rho \ge \eta \ge \epsilon$. However, we would not formulate this as a separate assumption but as choosing $\eta$ appropriately in the statement of Theorem 4.7. The reasoning is as follows:
> > >
> > > Suppose Assumption 3.1 holds with some constant $\ell > 0$. Then, by the argument in Appendix B.1 of [4], there exist constants $\ell' > 0$ and $\rho' > 0$ for which the extended MFCQ condition holds. While we might not know this $\rho'$, the proof in [4] gives a lower bound $\tilde{\rho} \leq \rho'$ which is a function of $\ell$, Lipschitz constants, and diameter of the strategy space. As we would assume to have bounds on these problem parameters, we can choose $\eta = \min(\epsilon,\tilde\rho)$. This implies that we satisfy $\rho' \ge \eta \ge \epsilon$, and hence our Theorem 4.7 gives the stated convergence guarantee with rate $\mathcal O(\epsilon^{-3})$. Note that the rate is unaffected by this adjusted choice of $\eta$, because if $\tilde\rho < \epsilon$, the iteration number is constant w.r.t. $\epsilon$, so this case does not change the $\mathcal O(\cdot)$-bound.

---

### Official Review · Reviewer_L5pB · 2026-03-13

**Soundness:** 3
**Presentation:** 3
**Significance:** 3
**Originality:** 3
**Overall Recommendation:** 5
**Confidence:** 3

**Summary:**

This paper considers concave games with shared coupled constraints under the assumption of player wise concave utilities and player wise concave constraints (which yields player wise convex feasible strategy sets). The paper makes two major theoretical contributions: 1) The paper proves the existence of constrained Nash equilibrium in such games. This is a non-trivial result as it does not assume the feasible strategy space to be jointly convex or the feasible response set to be non-empty. 2) For potential games lying in the above class of concave games, the paper uses log barrier regularized gradient descent under an extended MFCQ-like CQ condition to yield $\epsilon$-approximate Nash equilibrium within $\mathcal{O}(\epsilon^{-3})$ rounds for Lipschitz and smooth utilities/constraints.  Further, the paper presents empirical results of this algorithm for cooperative and network linked potential games.

**Compliance With Llm Reviewing Policy:**

Affirmed.

**Final Justification:**

The authors have addressed my questions and I maintain a positive opinion of the paper

**Key Questions For Authors:**

- I am curious about Assumption 4.5. It looks to me that this is required so that the game keeps getting driven into the interior of the feasible strategy space if it approaches the boundary. Is this assumption required only because you propose a log barrier based method, or is this assumption required due to some more fundamental issue that rise up because the game only assumes player wise concavity? In other words, if one was employing a log barrier technique on a game with a jointly convex feasible set, would this assumption still be required?

- Complete nitpick - but I believe the authors should use the word Generalized Nash equilibrium to better communicate the setting of coupled constraints. Please correct me if something in the setting prohibits the authors from doing so

**Limitations:**

yes

**Strengths And Weaknesses:**

I quite appreciate the removal of the need for the feasible region to be jointly convex in order to guarantee existence of Nash equilibria. This is quite a fundamental result that I am sure quite a lot of game settings will benefit from in the future.

---

> ### Author Rebuttal · Authors · 2026-03-31
>
> We appreciate the reviewer's positive feedback on our work and address their questions below.
> 1. Assumption 4.5 is indeed tied to the use of the log barrier method and ensures that the gradients push the iterates away from the boundary when getting close to it. A single-player analogue of this condition also appears in log barrier methods for nonconvex constrained optimization [1]. However, after receiving several remarks from reviewers about Assumption 4.5, we have revisited our result and related literature: It turns out that in our particular setting with compact strategy spaces and Lipschitz continuous constraint functions, this extended MFCQ condition is actually directly implied by Assumption 3.1, the non-extended version of our MFCQ condition used in our existence result (see also response to reviewer `7ciu`). The proof of this implication is contained in Appendix B.1 of [2]. Therefore, while the initial result under Assumption 4.5 was correct, for the final version of the paper, we will replace it with the strengthened version that relies only on the more common Assumption 3.1. We believe this improves the clarity and applicability of our result.
>
> 2. Regarding the notion of generalized Nash equilibrium problems (GNEPs), yes, there exist many variations in the literature of both GNEPs, as well as games with coupling constraints. We could not find a clear distinction between the two in terms of the problem structure; they appear to be used interchangeably. In our reading, “GNEP” is more common in the variational inequality literature, whereas “games with coupling constraints” is more prevalent in recent learning-oriented works closer to our setting. We agree that clarifying these connections may be helpful to the reader and will add a remark in the final version of the paper.
>
> ---
>
> [1] Usmanova, Ilnura, Yarden As, Maryam Kamgarpour, and Andreas Krause. "Log barriers for safe black-box optimization with application to safe reinforcement learning." Journal of Machine Learning Research 25, no. 171 (2024): 1-54.
>
> [2] Ni, Tingting, and Maryam Kamgarpour. "A Safe Exploration Approach to Constrained Markov Decision Processes." In International Conference on Artificial Intelligence and Statistics, pp. 3592-3600. PMLR, 2025.

---

> > ### Author Rebuttal · Reviewer_L5pB · 2026-04-03
> >
> > The authors have addressed my questions and I maintain a positive opinion of the paper.

---

### Official Review · Reviewer_RXzG · 2026-03-15

**Soundness:** 3
**Presentation:** 3
**Significance:** 2
**Originality:** 3
**Overall Recommendation:** 5
**Confidence:** 3

**Summary:**

The paper makes a contribution to the study of the existence and convergence to the Nash equilibrium in games where the set of admissible strategies is constrained. The authors show that even in the presence of shared constraints, under suitable technical requirements, there exists a player-decoupled algorithm that converges to equilibrium. Indeed, in the absence of shared constraints, in this setup, the Nash equilibrium does not have to exist.

**Compliance With Llm Reviewing Policy:**

Affirmed.

**Final Justification:**

I am now satisfied the paper assumptions are not as restrictive as I initially misunderstood.

**Key Questions For Authors:**

To what extent can Assumption 3.1 be satisfied in other games? Can you please provide some examples? Can you comment on the ability to extrapolate to other classes of games? What would be the main challenges?

**Limitations:**

I have a major concern regarding the generalizability and applicability of Assumption 3.1, which feels a bit artificial to me. If i read it right, the joint constraint set has a direction (the same for all agents) through which ALL agents can move towards the interior. This assumption should be checked explicitly in the two examples. The network routing game looks reasonable, but the cooperative game again feels artificial. Generally speaking, the range of experiments is limited, and more could be provided in the Appendix. I am willing to amend my score if I'm shown that the assumption is indeed somewhat common / reasonable and useful in other settings.

**Strengths And Weaknesses:**

The paper extends the literature on the topic. Exposition is clear and Example 3.2 is illustrative. I also appreciate the final remark, which gives intuition on the rate of convergence and places it in the context of other works.

---

> ### Author Rebuttal · Authors · 2026-03-31
>
> We thank the reviewer for their positive feedback and would like to make some clarifications regarding the concerns raised about Assumption 3.1.
>
> Assumption 3.1 does not require a common direction for all players. Rather, for each player $i$, there must exist a direction along which that player can move towards the interior of the feasible region ($\nabla_{x_i}c_j(x)$ refers to the gradient of the $j$-th constraint w.r.t. the $i$-th player's strategy). In this sense, the condition is a natural multi-player analogue of the classical MFCQ assumption from constrained optimization.
>
> For example, in the cooperative game from our simulations section, boundary points satisfy $x_1 x_2 = 0.15$. Thus for each player, decreasing their own variable (i.e., going in direction $-1$) moves the joint strategy towards the interior, so Assumption 3.1 holds.
>
> More broadly, assumptions of this type are commonly made in constrained game formulations. For example, [1] studies generalized Nash equilibrium problems with shared constraints under similar regularity conditions and includes a broad range of applications such as electricity markets, transportation networks, and spam filtering.
>
> ---
> [1] Dreves, Axel, Francisco Facchinei, Christian Kanzow, and Simone Sagratella. "On the solution of the KKT conditions of generalized Nash equilibrium problems." SIAM Journal on Optimization 21, no. 3 (2011): 1082-1108.

---

> > ### Author Rebuttal · Reviewer_RXzG · 2026-04-03
> >
> > I am now convinced Assumption 3.1 is not as restrictive as I initially though and have updated the score.

---

### Decision · Program_Chairs · 2026-04-30

**Decision:**

Accept (spotlight)

**Comment:**

This paper makes a meaningful and fundamental theoretical contribution to games with shared coupling constraints.  They show the existence of NE under playerwise concavity of utilities and constraints, without joint convexity of the feasible region. The proof strategy appears to be novel and interesting.  The paper also provides a convergence result for a independent learning algorithm in constrained potential games.